# The Role of Randomness in Stability

**Max Hopkins** [* 1]   **Shay Moran** [* 2 3]

## Abstract

Stability is a central property in learning and statistics promising the output of an algorithm $\mathcal{A}$ does not change substantially when applied to similar datasets $S$ and $S'$. It is an elementary fact that any sufficiently stable algorithm (e.g. one returning the same result with high probability, satisfying privacy guarantees, etc.) must be randomized. This raises a natural question: can we quantify *how much* randomness is needed for algorithmic stability? We study the randomness complexity of two influential notions of stability in learning: *replicability* (which promises $\mathcal{A}$ usually outputs the same result when run over samples from the same distribution), and *differential privacy* (which promises the output distribution of $\mathcal{A}$ remains similar under neighboring datasets). In particular, building on the ideas of (Dixon, Pavan, Vander Woude, and Vinodchandran ICML 2024) and (Cannone, Su, and Vadhan ITCS 2024), we prove a "weak-to-strong" boosting theorem for stability in these settings: the randomness complexity of a task $\mathcal{M}$ is tightly controlled by the best replication probability of any *deterministic* algorithm solving $\mathcal{M}$, a parameter known as $\mathcal{M}$'s "global stability" (Chase, Moran, Yehudayoff FOCS 2023). Finally, we use this connection to characterize the randomness complexity of PAC Learning: a class has bounded randomness complexity iff it has finite Littlestone dimension, and moreover scales at worst logarithmically in the excess error of the learner. As a corollary, we resolve a question of (Chase, Chornomaz, Moran, and Yehudayoff STOC 2024) about the error-dependent list-replicability of agnostic learning.

## 1. Introduction

Stability is a central property in learning and statistics promising the output of an algorithm $\mathcal{A}$ remains similar when applied to similar datasets $S$ and $S'$. It is an elementary fact that any sufficiently stable algorithm (e.g. one returning the same result with high probability, or one satisfying differential privacy) is either randomized, or constant.[1] This raises a natural question: *how much* randomness is needed for algorithmic stability?

In this work, we study the role of randomness in two fundamental notions of stability: replicability, and differential privacy. Replicability is the core scientific and algorithmic tenet that an experiment, run twice on data from the same underlying distribution, should produce the same output with high probability. Given a statistical task $\mathcal{M}$ and an algorithm for the task $\mathcal{A}$ (e.g. hypothesis selection, classification), one might reasonably define the replicability of $\mathcal{A}$ to be its (worst-case) collision probability over independent samples, that is:

$$\min_D \left\{ \Pr_{S \sim D^n} [\mathcal{A}(S) = \mathcal{A}(S')] \right\}$$

where $D$ ranges over the possible data distributions of $\mathcal{M}$. To achieve true replicability, we'd hope to construct an algorithm with collision probability near $1$. Unfortunately, it turns out this is unachievable. The best such parameter, called the *global stability* of $\mathcal{M}$ (Bun et al., 2020), is universally capped at $\frac{1}{2}$ for any non-trivial statistical task (Chase et al., 2023). Motivated by this fact, (Impagliazzo et al., 2022) recently proposed an elegant relaxation of global stability allowing *shared internal randomness*, calling an algorithm $\mathcal{A}$ $\rho$-*replicable* if over independent samples $S, S'$ and a shared random string $r$:

$$\forall D : \Pr_{S, S' \sim D^n, r} [\mathcal{A}(S; r) = \mathcal{A}(S'; r)] > 1 - \rho.$$

Global stability is then exactly the best replicability parameter achieved by any *deterministic* algorithm solving $\mathcal{M}$.[2]

---

[1]Why? This depends on the model of stability. A deterministic DP algorithm, for instance, must have exactly the same output over all neighboring samples. Since one can always construct a path of neighbors between any inputs $S$ and $S'$, $\mathcal{A}$ is inherently constant. In replicability a similar argument is possible but one interpolates between underlying distributions rather than samples.

[2]Traditionally a globally stable algorithm isn't assumed to be

*Equal contribution  [1]Department of Computer Science, Princeton University, Princeton, USA [2]Faculty of Mathematics, Technion, Haifa, Israel [3]Google Research, Tel-Aviv, Israel. Correspondence to: Max Hopkins <mh4067@princeton.edu>, Shay Moran <smoran@technion.ac.il>.

*Proceedings of the 42nd International Conference on Machine Learning*, Vancouver, Canada. PMLR 267, 2025. Copyright 2025 by the author(s).

In (Dixon et al., 2023), the authors introduce *certificate complexity*, the smallest number of random bits needed to achieve $\rho$-replicability. They prove tight bounds on the certificate complexity of several basic $d$-dimensional tasks, as well as showing near-matching bounds on the global stability.[3] Intuitively, it is reasonable to think there should be a general connection between the certificate complexity of a task and its global stability — the better the global stability, the better replication we can achieve using *no* random bits, so the easier it should be to achieve any particular replication threshold $\rho$. Our first main result confirms this intuition, proving a sort of 'weak-to-strong' boosting theorem for replicability: the number of random bits needed to beat $\frac{1}{2}$-replication probability is (up to a single bit) exactly inverse log of the global stability. Furthermore, as one might suspect, this can be amplified to any $\rho$-replicability using an additional $\log(1/\rho)$ random bits.

While replicability and global stability are important notions in their own right, they have perhaps been most impactful in their close relation to the widely influential notion of *differential privacy* (Dwork et al., 2006b). An algorithm for a statistical task is called differentially private if its output distribution remains similar under any two neighboring datasets. Like replicability, differentially private algorithms are inherently randomized, leading to the analogous notion of *DP complexity* (Canonne et al., 2024) measuring the smallest number of internal random bits required to achieve privacy. Our second main result is an analogous 'weak-to-strong' boosting theorem for differential privacy, closely tying DP Complexity to a task's underlying global stability. However, due to privacy's multiple parameters and nuanced dependence on sample complexity, our results in this context cannot be said to give an exact equivalence as above, and it remains an interesting open problem to fully characterize the relation between them.

Boosting in hand, we turn to look at the randomness complexity of one of the best studied tasks in learning: binary classification. We focus on the classical PAC model (Vapnik & Chervonenkis, 1974; Valiant, 1984), where a learner, given sample access to a distribution of labeled examples $D$, must produce a labeling from some fixed class $H$ that is close to the best possible option with high probability. The global stability of PAC learning is quite well studied (Bun et al., 2020; Ghazi et al., 2021a; Bun et al., 2023; Chase et al., 2023). In the 'realizable setting', where the data is assumed to be consistent with some $h \in H$, it is known how to construct a $2^{-2^{O(d)}}$-globally stable algorithm *independent of the learner's error*, for $d$ the Littlestone dimension

---

deterministic, but one can always de-randomize such an algorithm and achieve the same collision probability. See Appendix B.

[3]Formally, (Dixon et al., 2023) study a slightly different parameter called list-replicability, which is essentially equivalent to global stability (Chase et al., 2023).

of the class. Recently, (Chase et al., 2024) proved this result does not extend to the general 'agnostic' setting and ask whether it is instead possible to prove a bound that decays with the excess error $\alpha$. We resolve this problem: a class $H$ has bounded error-dependent global stability (and therefore also certificate complexity) if and only if it has finite Littlestone dimension. Moreover, the randomness complexity suffers only mild dependence on the error, scaling at worst as $O(\log \frac{1}{\alpha})$.

## 2. Main Results

We briefly overview our main setting of study. A **statistical task** $\mathcal{M}$ consists of a *data domain* $\mathcal{X}$, an *output domain* $\mathcal{Y}$, and, for every distribution $D$ over $\mathcal{X}$, a family $G_D \subset \mathcal{Y}$ of 'accepted solutions' for the task. An algorithm $\mathcal{A} : \mathcal{X}^* \to \mathcal{Y}$ 'solves' the task $\mathcal{M}$ if for any $\beta > 0$, given sufficiently many samples $n(\beta)$ from any $D$ over $\mathcal{X}$, $\mathcal{A}$ outputs $y \in G_D$ with probability at least $1 - \beta$. At a parametrized level, for fixed $\beta$ and $n(\beta)$, we say the algorithm solves $\mathcal{M}$ with *confidence* $\beta$ and *sample complexity* $n(\beta)$.

A randomized algorithm $\mathcal{A} : \mathcal{X}^* \times \{0,1\}^* \to \mathcal{Y}$ is called $\rho$-**replicable** if, for all large enough $n, \ell \geq 0$, over two independent samples $S, S' \sim D^n$ and the same shared random string $r \sim \{0,1\}^\ell$, $\mathcal{A}$ returns the same output with high probability:

$$\forall D : \Pr_{r \sim \{0,1\}^m, S, S' \sim D^n}[\mathcal{A}(S; r) = \mathcal{A}(S'; r)] \geq 1 - \rho.$$

The (parameter-free) **certificate complexity** of a task $\mathcal{M}$, denoted $C_{\text{Rep}}$, is the smallest $\ell$ such that there exists a better than $\frac{1}{2}$-replicable algorithm solving $\mathcal{M}$ *using at most $\ell$ random bits*, or, more explicitly, the smallest $\ell$ such that for every $\beta > 0$ there exists $n(\beta) \in \mathbb{N}$ and a better than $\frac{1}{2}$-replicable algorithm on $n(\beta)$-samples and $\ell$ random bits solving $\mathcal{M}$ with confidence $\beta$ (c.f. Definition A.4).

The **global stability** of a task $\mathcal{M}$ is the best replicability that can be achieved using *no* internal randomness. In particular, an algorithm $\mathcal{A}$ is called $\eta$-globally stable if over independent samples $S, S'$:

$$\forall D : \Pr_{S, S' \sim D^n}[\mathcal{A}(S) = \mathcal{A}(S')] \geq \eta.$$

The **globally-stable complexity** (g-stable complexity), denoted $C_{\text{Glob}}$, is $\log \frac{1}{\eta_{\mathcal{M}}}$, where $\eta_{\mathcal{M}} > 0$ is the supremum across $\eta$ for which there is a deterministic $\eta$-globally stable algorithm solving $\mathcal{M}$ (c.f. Definition A.2). Note that any $\eta$-globally stable algorithm automatically has an output which is an $\eta$-**heavy-hitter**, that is some $y \in \mathcal{Y}$ for which $\Pr[\mathcal{A}(S) = y] \geq \eta$. The traditional notion of global stability (Bun et al., 2020) only requires this latter condition. In Appendix B we show our variant is equivalent up to minor differences in sample complexity (meaning in particular the

g-stable complexity *does not depend* on which definition you take).

With this in mind, our first main result is a weak-to-strong boosting lemma for replicability: the number of random bits needed to achieve high probability replicability is exactly controlled by the best replication probability of any deterministic algorithm, i.e. g-stable and certificate complexity are (essentially) equivalent.

**Theorem 2.1** (Stability vs Replicability (Theorem C.1))**.** *Let $\mathcal{M}$ be any statistical task. Then:*

$$C_{Glob} \leq C_{Rep} \leq C_{Glob} + 1.$$

*Moreover, the number of random bits required to achieve $\rho$-replicability is at most $\lceil C_{Glob} + \log(1/\rho) \rceil$.*

Replicability and global stability have played an important role in recent work constructing efficient differentially private algorithms in learning and statistics (Bun et al., 2020; Ghazi et al., 2021b; Bun et al., 2023; Kalavasis et al., 2023). An algorithm $\mathcal{A}$ is said to be $(\varepsilon, \delta)$-**differentially private** if for any two datasets $S, S' \in \mathcal{X}^n$ differing in only one coordinate, the output distribution of $\mathcal{A}$ is nearly indistinguishable in the following sense. For any measurable event $\mathcal{O} \subset \mathcal{Y}$:

$$\Pr[\mathcal{A}(S) \in \mathcal{O}] \leq e^{\varepsilon} \Pr[\mathcal{A}(S') \in \mathcal{O}] + \delta.$$

In recent work, (Canonne et al., 2024) introduce the *DP complexity* of a task $\mathcal{M}$, denoted $C_{DP}(n, \beta, \varepsilon, \delta)$, measuring the minimum number of random bits required to construct an $(\varepsilon, \delta)$-DP algorithm on $n$ samples solving $\mathcal{M}$ with $\beta$-confidence. Unlike our prior notions, DP complexity is *parametrized* due to the fact that there is no clear 'threshold' to set for the privacy parameters $(\varepsilon, \delta)$ as in the replicability setting (i.e. $\rho = 1/2$). Furthermore, $(\varepsilon, \delta)$-privacy is only meaningful when the parameters are taken as functions of the sample complexity $n$ (and therefore confidence $\beta$), leading to the somewhat cumbersome $C_{DP}(n, \beta, \varepsilon, \delta)$.

As a result of the above, the connection between DP complexity and global stability is somewhat more nuanced than the parameter-free equivalence in Theorem 2.1. We will consider two variants of the connection. In the first, we compare DP complexity to analogously defined parametrized g-stable and certificate complexities $C_{Glob}(n, \beta)$ and $C_{Rep}(n, \beta)$ (see Appendix A.2 for formal definition).

**Theorem 2.2** (Stability vs DP (Informal Theorem D.1))**.** *There exists a universal constant $c > 0$ such that for any statistical task $\mathcal{M}$:*

1. *(**Stability to DP**): $C_{DP}(n, \beta, \varepsilon, \delta) \leq C_{Glob}(n', \beta') + \log(1/\varepsilon) + \log(1/\delta) + \tilde{O}(1)$*

   *for any $n \geq n' \cdot \exp(C_{Glob}(n', \beta')) \frac{\log(\frac{1}{\delta}) \log(\frac{1}{\beta})}{\varepsilon}$*

   *and $\beta \geq \beta' \cdot \exp(C_{Glob}(n', \beta')) \frac{\log(\frac{1}{\delta})}{\varepsilon}$*

2. *(**DP to Stability**): $C_{Glob}(n, \beta) \leq C_{DP}(n', \beta', \varepsilon, \delta) + O(1)$,*

*for any $(n', \varepsilon, \delta)$ satisfying $\varepsilon \leq \frac{c}{\sqrt{n' \log(n')}}$ and $\delta \leq \frac{c}{n'}$, $n \geq n' \exp(C_{Glob}(n, \beta))$, and $\beta \geq \beta' \exp(C_{Glob}(n, \beta))$*

Qualitatively, Theorem 2.2 (Item 1) simply states that given an $\eta$-globally stable (deterministic) algorithm, we can 'boost' it into an $(\varepsilon, \delta)$-DP algorithm using only $\log(1/\eta) + \log(1/\varepsilon) + \log(1/\delta)$-random bits, where the new algorithm may have somewhat higher failure probability ($\beta$ vs $\beta'$) and sample complexity ($n$ vs $n'$) than the original. This should be compared to the analogous statement in replicability (Theorem 2.1), which roughly stated an $\eta$-globally stable algorithm can be boosted to a $\rho$-replicable one using $\log(1/\eta) + \log(1/\rho)$ random bits. Theorem 2.2 (Item 2) states the (weak) converse also holds: given a good enough DP algorithm on $\log(1/\eta)$ random bits, we can 'reverse-engineer' a deterministic $\eta$-globally stable algorithm from it, again paying in the sample size and confidence.

We remark the blowup in sample complexity and confidence in Theorem 2.2 is typically mild — while the exponential dependence on $C_{\text{Glob}}$ looks large, $C_{\text{Glob}}$ is log-scale, so the blowup is only polynomial in stability similar to prior works in the area (Bun et al., 2020; 2023). Furthermore, almost all learning algorithms have logarithmic sample dependence on the failure probability $\beta$ (indeed in most cases one can amplify a constant success algorithm to confidence $\beta$ simply by repeating it $\log(1/\beta)$ times—in this sense the loss in confidence only results in paying a $\log \log$ factor in the global stability. Nevertheless, we emphasize the core focus of our work is on one specific computational resource—randomness. As a result, we sometimes compromise on other resources (as we do here) to better understand the fundamental limits of the main one.

Nevertheless, since the quantitative statement of Theorem 2.2 is fairly cumbersome otherwise, it is still worth taking a minute to explain the various dependencies before moving on. Recall our goal: given an $\eta$-globally stable algorithm $\mathcal{A}_{\text{global}}$ on $n'$ samples that succeeds with probability $1 - \beta'$, we wish to use $\mathcal{A}_{\text{global}}$ as a subroutine to construct an $(\varepsilon, \delta)$-DP algorithm solving the same problem with slightly worse probability $1 - \beta$ and using slightly more $(n)$ samples. As in prior work, the key to doing this is to build a large database of $\mathcal{A}_{\text{global}}$'s $\eta$-*heavy-hitters*, outputs of $\mathcal{A}_{\text{global}}$ that appear with probability at least $\eta$. By standard concentration bounds, this can be done by running $\mathcal{A}_{\text{global}}$ roughly $O\left(\frac{\log \frac{1}{\beta}}{\eta^2}\right)$ times and looking at empirical output probabilities. Outputting one of these heavy-hitters privately actually requires us to run this estimation process $O\left(\frac{\log \frac{1}{\delta}}{\eta \varepsilon}\right)$ times to ensure the resulting 'dataset' of heavy hitters isn't too strongly affected by changing any single input sample in the

process. In total, the above uses $n = n' \cdot \frac{\log(\frac{1}{\delta})\log(\frac{1}{\beta})}{\eta^3 \varepsilon}$ total samples and setting $\eta$ to be the best achievable replication probability $(2^{-C_{\text{Glob}}(n', \beta')})$ gives the stated dependence. The decay of the success probability $\beta$ can similarly be thought of as coming from bounding the probability any individual subroutine $\mathcal{A}_{\text{global}}$ fails, though we give a better bound through more careful analysis in the main body.

Of course the basic procedure described above, which is similar to prior DP-to-stability reductions in the literature (Bun et al., 2023), is not randomness-efficient (indeed it may even use an unbounded number of random bits!) As we discuss in Section 4.2, our randomness-efficient variant relies on carefully discretizing this type of transform to optimize the number of random bits without sacrificing correctness and privacy.

In the reverse direction (Item 2), we directly prove any sufficiently private algorithm (i.e. one with the stated dependencies on $(n, \varepsilon, \delta)$) that uses $k$ random bits *automatically* has a heavy-hitter of weight roughly $\exp(-k)$. We then show how to transform any $\beta'$-correct randomized algorithm $\mathcal{A}$ on $n'$ samples with an $\eta$-heavy-hitter into a deterministic $\eta$-globally stable algorithm at the cost of running $\mathcal{A}$ roughly $\frac{\log(1/\beta)}{\eta^2}$ times, resulting in similar parameter blowups as in Item 1. We remark that the constraints on $(n, \varepsilon, \delta)$ are also in general fairly mild. The assumption on $\delta$ is extremely weak (an algorithm is not considered private unless $\delta \leq n^{-\omega(1)}$). The assumption on $\varepsilon$ is more restrictive, but is satisfied by many basic DP mechanisms, and, moreover, it is almost always possible to amplify a weak DP algorithm to one satisfying this constraint (though it may cost many additional random bits). We discuss this further in Section 3.

Moving on from Theorem 2.2, we'd also like a way to compare randomness complexity in DP to our original *parameter-free* variants of $C_{\text{Glob}}$ and $C_{\text{Rep}}$. It turns out this is possible by considering a slight generalization of vanilla differential privacy called *user-level* DP. In user-level DP there are $T$ 'users', each of whom contributes a (sub)-dataset $S_i$. Neighboring datasets are defined with respect to swapping an entire user rather than a single example. In other words, in user-level DP we view the total size-$n$ dataset $S$ as being comprised of $T$ components ('users') $(S_1, \ldots, S_T) \in (\mathcal{X}^{n/T})^T$, and must maintain $(\varepsilon, \delta)$-privacy under swapping out an entire $S_i$ subsample. Critically, in this setting $(\varepsilon, \delta)$ are now *functions of $T$* rather than of $n$. This allows us to define the user-level DP complexity, $C_{DP}(T, \varepsilon, \delta)$, in a way that is independent of sample complexity and confidence as the smallest number of random bits such that there exists a $T$-user $(\varepsilon, \delta)$-DP algorithm solving $\mathcal{M}$. We then get the following cleaner 'parameter-free' version of Theorem 2.2:

**Theorem 2.3** (Stability vs User-Level DP (Informal Theorem D.2)). *There exist universal constants $c_1, c_2 > 0$ such that for any statistical task $\mathcal{M}$:*

1. *(**Stability to DP**):* $C_{DP}\left(2^{C_{\text{Glob}}\frac{c_1 \log \frac{1}{\delta}}{\varepsilon}}, \varepsilon, \delta\right) \leq C_{\text{Glob}} + \log(1/\varepsilon) + \log(1/\delta) + \tilde{O}(1).$

2. *(**DP to Stability**):* $C_{\text{Glob}} \leq C_{DP}(T, \varepsilon, \delta) + O(1),$

*where the latter holds for any $(T, \varepsilon, \delta)$ satisfying $\varepsilon \leq \frac{c_2}{\sqrt{T \log(T)}}$ and $\delta \leq \frac{c_2}{T}$.*

We remark that the parameter dependencies here follow exactly the same explanation as in Theorem 2.2, with $T = 2^{C_{\text{Glob}}\frac{c_1 \log \frac{1}{\delta}}{\varepsilon}}$ in (Item 1) appearing due to finding heavy hitters of $\mathcal{A}_{\text{global}}$, and the $(T, \varepsilon, \delta)$ dependency in (Item 2) needed to imply the corresponding DP algorithm has a heavy hitter.

### 2.0.1. THE STABLE COMPLEXITY OF PAC LEARNING

Having established our stability boosting theorems, we turn to the randomness complexity of binary classification. We focus on the standard **PAC Learning** model (Valiant, 1984; Vapnik & Chervonenkis, 1971). A PAC Learning classification problem consists of a *data domain* $X$ and a *hypothesis class* $H = \{h : X \to \{0, 1\}\}$ of potential labelings of $\mathcal{X}$. Given a distribution $D$ over labeled samples $X \times \{0, 1\}$, the *classification error* of a hypothesis $h$ is

$$err_D(h) := \Pr_{(x,y)\sim D}[h(x) \neq y].$$

An algorithm is said to (agnostically) PAC learn the class $(X, H)$ if for every $\alpha, \beta > 0$ there exists $n = n(\alpha, \beta) \in \mathbb{N}$ such that given $n$ samples, $\mathcal{A}$ outputs an $\alpha$-optimal hypothesis with probability at least $1 - \beta$:

$$\forall D : \Pr_{S\sim D^n}[err_D(\mathcal{A}(S)) > \min_{h \in H} err_D(h) + \alpha] \leq \beta.$$

In our framework, PAC Learning can be viewed as a sequence of statistical tasks $\{\mathcal{M}_\alpha\}$ parameterized by the error $\alpha$, where $\mathcal{X} = X \times \{0, 1\}$, $\mathcal{Y}$ is the set of all labelings of $X$, and $G_D$ is the set of $\alpha$-optimal labelings.

Due to its close connection with differential privacy, stability is quite well studied in the PAC setting. In (Bun et al., 2020), the authors show that under the assumption that $\min_{h \in H} err_D(h) = 0$ (called the 'realizable setting'), the g-stable complexity of $\{\mathcal{M}_\alpha\}$ can be universally upper bounded by $2^{O(d)}$ for any class $(X, H)$ with finite Littlestone dimension $d$. In the error-dependent setting, the best known bound is of (Ghazi et al., 2021a;b) who improve the g-stable complexity to $\text{poly}(d) + O(\log(\frac{1}{\alpha}))$.

Contrary to the above, (Chase et al., 2024) show the former type of error-independent bound is impossible in the agnostic setting, and ask whether an error-dependent bound

like (Ghazi et al., 2021a) can be extended to this case.[4] We resolve this problem: not only is such a bound possible, the complexity scales essentially as in the realizable setting up to a factor in the VC dimension.

**Theorem 2.4** (The Certificate Complexity of Agnostic Learning (Theorem E.1)). *Let $(X, H)$ be a hypothesis class with Littlestone dimension d. Then $(X, H)$ has a better than $\frac{1}{2}$-replicable learner with*

1. ***Sample Complexity:***

$$\exp(\text{poly}(d)) \, \text{poly}(\alpha^{-1}, \log(1/\beta))$$

2. ***Certificate Complexity:***

$$\text{poly}(d) + O(VC(H) \log(\frac{1}{\alpha}))$$

*Conversely if $d = \infty$, then $C_{Rep}(\alpha) = \infty$ for any $\alpha < \frac{1}{2}$. In other words, there is no globally stable, DP, or replicable algorithm for $(X, H)$ better than random guessing.*

While we defer the definition of Littlestone dimension itself to Appendix A (the notion is not needed for our arguments which use prior work to this end as a black box), it is useful to mention a few concrete examples for intuition. For instance, the class of $k$-dimensional affine subspaces of $\mathbb{R}^d$ has Littlestone dimension $k + 1$, and the class of $\gamma$-margin halfspaces in $\mathbb{R}^d$ has Littlestone dimension at most $\frac{1}{\gamma^2}$. In both cases Theorem 2.4 gives the first randomness-efficient replicable algorithm for learning these classes (and in fact the first globally stable algorithm for the problem at all).

We remark if all one wants is heavy-hitter global stability, the sample complexity of Theorem 2.4 can be improved to $\text{poly}(d, \alpha^{-1}, \log(1/\beta))$ while maintaining $(\text{poly}(d) + O(VC(H) \log(\frac{1}{\alpha})))$-g-stable complexity. It is an interesting question whether $\text{poly}(d, \alpha^{-1}, \log(1/\beta))$ samples is achievable in the replicable case while maintaining good certificate complexity. This cannot be achieved using current sample-efficient methods which all rely on correlated sampling (Ghazi et al., 2021b; Bun et al., 2023) and have certificate size scaling with $|H|$. Simultaneously achieving randomness and sample efficiency therefore seems to require new ideas in the theory of replicable algorithm design.

In recent independent work, (Blonda et al., 2025) also prove a variant of Theorem 2.4 for global stability with $g$-stable complexity $\exp(d) + \text{poly}(\alpha^{-1})$. Their work also addresses a second open problem of (Chase et al., 2024) which we do not consider. See Section 3.1 for further discussion.

---

[4]Again, we note (Chase et al., 2024) is phrased in terms of 'list-replicability', but (Chase et al., 2023) prove the $\alpha$-dependent (log) list-size $C_{\text{List}}(\alpha)$ satisfies $C_{\text{Glob}}(\alpha) \le C_{\text{List}}(\alpha) \le C_{\text{Glob}}(\alpha/2)$, so it is asymptotically equivalent to characterize $C_{\text{Glob}}(\alpha)$. We refer the reader to Section 5 of (Chase et al., 2023) and Section 2.3.1 of (Chase et al., 2024) for details and the formal definitions.

## 3. Discussion and Open Problems

Before moving to the formal details, we take some space to further discuss the parameter restrictions in Theorem 2.2 and Theorem 2.3, related open problems, and provide some further justification why counting internal random bits is sensible for statistical tasks where 'external' randomness is also available via samples.

**DP Complexity and the Pareto Frontier:** Recall both Theorem 2.2 and Theorem 2.3 require a variant of the following guarantee on the privacy parameters $(T, \varepsilon, \delta)$

$$\varepsilon \lesssim \frac{1}{\sqrt{T \log(T)}}, \delta \lesssim \frac{1}{T} \tag{1}$$

where $T$ is the number of users (in the vanilla DP setting, $T = n$ is then just the number of samples). Our first main question is to what extent such a requirement is necessary. Can we characterize for what triples $(T, \varepsilon, \delta)$ the inequality $C_{\text{Glob}} \le C_{DP}(T, \varepsilon, \delta) + O(1)$ holds?

Toward this end, it is worth briefly discussing why this constraint occurs, and to what extent it is normally achievable. Regarding the first, Equation (1) are exactly the constraints required to achieve a strong notion of stability called *perfect generalization* (Cummings et al., 2016; Bassily & Freund, 2016), which promises that on most input datasets $S$, $\mathcal{A}(S; \cdot)$ is actually statistically close to the distribution $\mathcal{A}(\cdot; \cdot)$. All known transforms between DP and replicability go through perfect generalization (Bun et al., 2023; Kalavasis et al., 2023) (we note prior transforms combine this with a method called correlated sampling which ruins the certificate size, so do not recover our results). At a high level, it would be interesting to give a transform between DP and list-replicability without going through perfect generalization, potentially bypassing this barrier. In some sense such a transformation is actually possible in the PAC Learning setting using the fact that Littlestone dimension characterizes learning, but this results in tower-type dependence[5] in the certificate size.

Regarding the achievability of Equation (1), in most cases such constraints are fairly mild. In fact, normally Equation (1) is hardly a barrier at all since it is always possible to *amplify* a $T$-user $(O(1), 1/\text{poly}(T))$-DP algorithm to a $\frac{T}{\varepsilon}$-user $(\varepsilon, 1/\text{poly}(T))$-DP algorithm simply by applying the former on a random size-$T$ sub-sample of the latter's users (Balle et al., 2018). Setting $\varepsilon \ll \frac{1}{T}$ then satisfies Equation (1). The catch here is that this process uses $T \log(1/\varepsilon)$ additional random bits, a heavy cost when known settings seem to only require *logarithmically* many bits in $T$ (Canonne et al., 2024). This leads to an interesting question: is it possible to derandomize subsampling techniques in differential privacy? Can we amplify

---

[5]I.e. a dependence of type $T(n) = 2^{T(n-1)}$.

an $(O(1), 1/\operatorname{poly}(T))$-DP algorithm to $(\varepsilon, 1/\operatorname{poly}(T))$-DP using only $O(\log(T/\varepsilon))$ additional random bits?

**Measuring Randomness in Statistics:** Certificate and DP complexity count the number of *internal* random bits used by an algorithm solving a fixed statistical task $\mathcal{M}$. Unlike the empirical setting (as studied e.g. in (Canonne et al., 2024)), algorithms in the statistical setting also have access to *external* randomness in the form of i.i.d samples from the underlying data distribution $D$. It is reasonable to then ask: why do such algorithms need internal randomness at all, and (assuming they do) why does it make sense to measure it?

For certificate complexity, this question was naturally addressed in the seminal works of (Impagliazzo et al., 2022; Dixon et al., 2023). Recall that in replicability, the internal randomness of $\mathcal{A}$ plays a special role apart from the sample because it is *shared* between the two runs of the algorithm. It is not hard to argue that most natural problems (namely any problem where there is a continuous path between two distributions $D$ and $D'$ with disjoint solution sets) require at least one shared random bit to achieve better than $1/2$ replication probability.[6] Another natural interpretation of the certificate complexity is as a measure of the 'communication complexity' of a statistical task — it is the number of bits that must be published publicly to ensure external parties can verify the result of the algorithm.

In differential privacy, the situation is more nuanced. In (Canonne et al., 2024), the authors focus on empirical tasks, meaning the dataset is fixed (not drawn i.i.d from some distribution) and the only source of randomness is internal. In this case it is obvious that any non-trivial DP algorithm must be randomized, and there is strong motivation to bound the extra number of random bits required since 1) clean randomness is expensive, and 2) prior implementations of DP in practice such as the U.S. Census used an astronomical number of clean random bits, thought to be upwards of 90 terabytes (Garfinkel & Leclerc, 2020). Understanding to what extent this cost can be mitigated is then a question of significant theoretical and practical interest.

In the statistical setting, we argue that despite our additional access to external randomness, any useful notion of differential privacy must still inherently rely only on internal randomness to protect user data. This is because differential privacy should always protect users in the *worst-case*, even for *average-case problems* studied in statistics. In particular, since the algorithm has no control over the sample

oracle, relying on its output to ensure privacy leaves it open to attacks corrupting the oracle, or even to privacy failure stemming from benign issues such as light failure of the i.i.d assumption in real-world data (see e.g. discussion in (Goldwasser et al., 2022) regarding possible breaches in security from corruption of a learning algorithm's randomness). To avoid such scenarios, when defining differential privacy for statistical tasks we always apply privacy at the level of the *fixed sample*, so the same motivation and context as in the empirical case holds.

### 3.1. Further Related Work

**Stability and Replicability:** Global stability was introduced in (Bun et al., 2020) to upper bound the sample complexity of private PAC Learning. Replicability was introduced independently in (Impagliazzo et al., 2022) and (Ghazi et al., 2021b) as a relaxation of global stability, in the former towards addressing the crisis of replicability in science, and in the latter towards achieving upper bounds on the sample complexity of user-level private learning. Since these works, a great deal of effort has gone into understanding what statistical tasks admit replicable algorithms (Karbasi et al., 2023; Esfandiari et al., 2024; Eaton et al., 2024; Esfandiari et al., 2022; Komiyama et al., 2024; Hopkins et al., 2024; Kalavasis et al., 2024), as well as their connection to other notions of stability such as differential privacy (Ghazi et al., 2021b; Bun et al., 2023; Kalavasis et al., 2023; Moran et al., 2023; Chase et al., 2023). Our work is most related to this latter line, especially Theorem 2.2 and Theorem 2.3 which rely on new randomness-efficient variants of the transforms introduced in these works.

Replicability is substantially stronger than classical stability notions in the literature (see e.g. (Bousquet & Elisseeff, 2002)) which generally require that similar inputs result in *close* rather than equivalent outputs. While this may seem too strong a guarantee to ask for in statistical analysis, it's worth highlighting that requiring true equivalence comes with a host of benefits not enjoyed by prior notions. This is discussed extensively in prior work (see e.g. (Impagliazzo et al., 2022; Bun et al., 2023)), and we will mention just a few such examples here:

1. Replicability is closely related to other 'strong' notions of stability in computer science (DP, adaptive data analysis, and more). Several open problems in DP (sample-efficient user-level DP, amplification of DP) were only resolved through studying replicability (Ghazi et al., 2021b; Bun et al., 2023).

2. Replicability is testable. It is not always easy to test whether an algorithm is stable (e.g. testing DP is known to be computationally hard (Gaboardi et al., 2020)). By requiring equivalence, replicability is trivially verifi-

---

[6]The argument is simply that if the algorithm is correct, it must be outputting different solutions on $D$ and $D'$, but then there is some interior distribution on the path which is non-replicable. See (Dixon et al., 2023; Impagliazzo et al., 2022; Chase et al., 2023) for more detailed examples.

able.

3. Replicability is preserved under arbitrary post-processing. Even applying a very sensitive function to the output of a replicable algorithm remains replicable, while such a procedure will not preserve weak stability.

4. Replicability is easily amplified. Given a constantly replicable algorithm, there is a simple and computationally efficient procedure to amplify it to arbitrarily high replication probability.

5. Replicability is model independent. Different statistical tasks naturally have different notions of weak stability and closeness, or, in the case of testing problems with binary outputs, may have no natural notion of weak stability (see (Hopkins et al., 2024) for further discussion). Replicability gives a unified definition for studying stability for all statistical tasks.

In general, it is interesting to understand the tradeoffs inherent between weaker classical notions and replicability. If one wants very high probability stability or is handling very high dimensional data, weaker notions will be computationally and statistically cheaper. On the other hand in low-dimensional settings ensuring strong replicability often has little to no asymptotic overhead (Impagliazzo et al., 2022), and when safety, testability, or privacy is paramount, it is necessary to use such a strong notion. Developing a better understanding of both is critical to help us decide when each should be used in practice.

**Certificate and DP Complexity:** Certificate complexity was introduced in (Dixon et al., 2023), where the authors prove essentially tight bounds for the basic task of estimating the bias of $d$ coins and learning classes via non-adaptive statistical queries. Related to our results, (Dixon et al., 2023) observe (at least implicitly) the basic connection that any $\rho$-replicable algorithm on $\ell(\rho)$ bits has a $2^{-\ell(\rho)}$-heavy-hitter, a basic component of the (replicability $\rightarrow$ global stability) direction of Theorem 2.1. They also use discretized rounding, which is the core component of the reverse direction, but focus in particular on $d$-dimensional space while we give a generic discretized rounding scheme based on access to a 'weakly replicable' subroutine over any domain.

**DP complexity** was introduced recently in (Canonne et al., 2024), where the authors study the number of random bits required to perform counting queries under $(\varepsilon, \delta)$-differential privacy on a fixed empirical dataset. The authors lower bound techniques use similar methods to those used to bound certificate complexity in (Dixon et al., 2023), but they do not give any generic connection between the two.

**Stability in Agnostic Learning:** In recent independent work, (Blonda et al., 2025) also resolve (Chase et al., 2024)'s question on error-dependent global stability in agnostic PAC-learning, proving a variant of Theorem 2.4 with randomness complexity $\exp(d) + \mathrm{poly}(\alpha^{-1})$. Beyond this, (Blonda et al., 2025) also prove impossibility for error-independent global stability when OPT is bounded above by some known constant $\gamma$, generalizing the error-independent stability impossibility result of (Chase et al., 2024). We do not consider the latter model in our work.

## 4. Technical Overview

In this section we overview of the main ideas in the proofs of Theorem 2.1, Theorem 2.2 and Theorem 2.3, and Theorem 2.4 respectively. We emphasize the below are not full proofs, and refer the reader to Appendix C, Appendix D, and Appendix E for the full details.

### 4.1. Global Stability and Certificate Complexity

We start with the forward direction: given a $(\frac{1}{2} + \gamma)$-replicable algorithm $\mathcal{A}_{\mathrm{Rep}}$ for $\mathcal{M}$ on $\ell = C_{\mathrm{Rep}}$ random bits, we transform it into a deterministic $\eta$-globally stable algorithm for $\mathcal{M}$ for any $\eta < 2^{-\ell}$. Since the g-stable complexity of $\mathcal{M}$ is the infimum of $\log \frac{1}{\eta}$ for all achievable $\eta$-globally stable algorithms, this implies $C_{\mathrm{Glob}} \leq C_{\mathrm{Rep}}$.

Our first step is to use $\mathcal{A}_{\mathrm{Rep}}$ to build a (randomized) algorithm with a nearly $2^{-\ell}$-heavy-hitter. We will then transform this algorithm into a deterministic one with nearly $2^{-\ell}$ replication probability. Toward this end, observe that by averaging for every distribution $D$ there exists a random string $r_D$ such that $\mathcal{A}_{\mathrm{Rep}}$ has a 'canonical solution' $h_D$ occurring with probability at least $\frac{1}{2} + \gamma$:

$$\Pr_{S \sim D^n}[\mathcal{A}_{\mathrm{Rep}}(S; r_D) = h_D] \geq 1/2 + \gamma.$$

By running $\mathcal{A}_{\mathrm{Rep}}(\cdot; r)$ roughly $\frac{\log(1/\tau)}{\gamma^2}$ times over fresh samples and taking the majority output, we therefore get an algorithm $\mathcal{A}_{\mathrm{maj}}$ with a nearly $2^{-\ell}$-heavy hitter

$$\Pr_{r \sim \{0,1\}^\ell, S \sim D^{n_2}}[\mathcal{A}_{\mathrm{maj}}(S; r) = h_D] \geq 2^{-\ell}(1 - \tau),$$

since $r = r_D$ with probability $2^{-\ell}$, and conditioned on this event $\mathcal{A}_{\mathrm{maj}}$ almost always outputs $h_D$ by Chernoff.

We now transform $\mathcal{A}_{\mathrm{maj}}$ into a *deterministic* algorithm with nearly $2^{-\ell}$ replication probability. This follows in two steps. First, we construct a new randomized algorithm $\mathcal{A}_{\mathrm{maj}}^{\mathrm{HH}}$ which runs $\mathcal{A}_{\mathrm{maj}}$ $2^{O(\ell)}$ times (over fresh samples and randomness) and output the most common response. Standard concentration then ensures the output of $\mathcal{A}_{\mathrm{maj}}^{\mathrm{HH}}(S; r)$ is a nearly $2^{-\ell}$-heavy-hitter of $\mathcal{A}_{\mathrm{maj}}$ with very high probability over $S$ and $r$. Second, we derandomize $\mathcal{A}_{\mathrm{maj}}^{\mathrm{HH}}(S; r)$ by taking $\mathcal{A}_{\mathrm{maj}}^{\mathrm{HH\text{-}det}}(S)$ to be the plurality response of $\mathcal{A}_{\mathrm{maj}}^{\mathrm{HH}}(S; r)$ over all random strings. A simple Markov-type argument

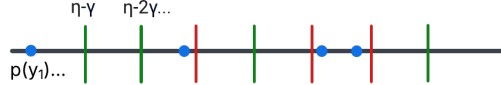

*Figure 1.* Thresholding procedure for $C_{\text{Glob}} = 2$ and $T = 7$. Blue dots denote the 4 heavy hitters, one of which $p(y_1)$ is known to be far from any threshold. This leaves 4 (green) thresholds with no nearby heavy-hitters out of 7, so $\rho \approx 4/7 > \frac{1}{2}$, and $C_{\text{Rep}} \leq 3$.

shows that $\mathcal{A}_{\text{maj}}^{\text{HH-det}}(S)$ also almost always outputs a nearly $2^{-\ell}$-heavy-hitter of $\mathcal{A}_{\text{maj}}$ over the randomness of $S$, and since there are at most $2^{\ell}$ such heavy hitters this implies $\mathcal{A}_{\text{maj}}^{\text{HH-det}}(S)$ is (nearly) $2^{-\ell}$-globally stable as desired.

We remark that correctness is maintained throughout all the above transforms so long as our initial confidence $\beta \leq 2^{-O(\ell)}$ was sufficiently small, since then any heavy hitters of the original algorithm must have been correct in the first place, and we almost always output such a hypothesis.

To prove the reverse direction, we need to show how to amplify an $\eta$-globally stable algorithm $\mathcal{A}_{\text{global}}$ to a $\rho$-replicable one using $\log \frac{1}{\eta} + \log \frac{1}{\rho}$ random bits. The key is again to look at the set of heavy-hitters of $\mathcal{A}_{\text{global}}$. For $y \in \mathcal{Y}$, let $p(y)$ denote the probability $\mathcal{A}_{\text{global}}$ outputs $y$. Running $\mathcal{A}_{\text{global}}$ sufficiently many times, we get empirical estimates $\hat{p}(y)$ such that $|\hat{p}(y) - p(y)| < \frac{\gamma}{3}$ with very high probability for some small $\gamma \ll \text{poly}(\rho\eta)$.

Fix $T \approx \frac{1}{\eta\rho}$ and consider the set of thresholds $\{\eta - \gamma, \eta - 2\gamma, \ldots, \eta - T\gamma\}$. Our final algorithm simply selects one of the $T$ thresholds $t$ at random, and outputs the first $y$ (with respect to some fixed order on $\mathcal{Y}$) satisfying $\hat{p}(y) \geq t$.

We now argue the above procedure is $\rho$-replicable. To see this, observe that because $\mathcal{A}_{\text{global}}$ only has $\frac{1}{\eta}$ heavy hitters of weight more than $\eta - T\gamma$ (and moreover one of weight at least $\eta$), it must be the case that at most $\frac{1}{\eta} - 1$ of these thresholds are within $\gamma/3$ of some $p(y)$ for any $y \in Y$. On the other hand, conditioned on the fact $|\hat{p}(y) - p(y)| < \gamma/3$, choosing any of the $T - \frac{1}{\eta} + 1$ thresholds $t$ without such a nearby heavy hitter results in a completely replicable output, since the list of $y$ with $\hat{p}(y) > t$ is always the same. The probability of selecting such a threshold is $\frac{T - \frac{1}{\eta} + 1}{T} > 1 - \rho$, so taking the failure probability of our empirical estimates sufficiently small gives a $\rho$-replicable algorithm as desired. See Figure 1 for an example diagram of this procedure.

Similar to before, correctness is also maintained as long as our original algorithm is $\beta \leq O(\eta)$ confident, as then every heavy hitter must be correct and the algorithm almost always outputs a heavy hitter of $\mathcal{A}_{\text{global}}$.

## 4.2. Global Stability and Differential Privacy

We now overview the main ideas behind Theorem 2.2 and Theorem 2.3 in more detail. We start with the forward direction: given an $\eta$-globally stable algorithm, we'd like to construct an $(\varepsilon, \delta)$-DP algorithm using roughly $\log(1/\eta) + \log(1/\varepsilon) + \log(1/\delta)$ random bits. The key to doing this is to first construct an $(\varepsilon, \delta/2)$-DP-algorithm $\mathcal{A}_{DP}$ satisfying the following properties

1. Bounded Support:

$$\forall S : |\text{Supp}(\mathcal{A}_{DP}(S))| \leq \tilde{O}\left(\frac{1}{\eta\varepsilon}\right),$$

2. Strong Correctness:

$$\Pr_S[\text{Supp}(\mathcal{A}_{DP}(S)) \subset G_D] \geq 1 - \beta.$$

In other words, the algorithm should *always* have small support, and that support should almost always be *completely* correct. The only issue is that $\mathcal{A}_{DP}$ might use too many random bits. We address this via an elegant observation of (Canonne et al., 2024): any distribution of support $T$ can be $\delta/2$-approximated (i.e. we can produce another distribution within $\delta/2$ in total variation distance) using only $\log(T) + \log(2/\delta)$ random bits and without adding any new elements to the support. Given the above conditions, this results in a $\beta$-confident $(\varepsilon, \delta)$-DP algorithm as desired.

Building on prior work, we construct our base DP algorithm using the 'DP Selection' primitive of (Korolova et al., 2009; Bun et al., 2016; 2018), which, roughly, given a dataset $M$, provides an $(\varepsilon, \delta)$-DP procedure to output some $y \in \mathcal{Y}$ that appears no fewer than $\frac{\log(1/\delta)}{\varepsilon}$ times less than the most frequent element in $M$. As discussed in the previous sections, given $\mathcal{A}_{\text{global}}$ we can build such a dataset $M$ by 1) building an algorithm that outputs one of $\mathcal{A}_{\text{global}}$'s heavy hitters with high probability 2) running this procedure $O(\frac{\log \frac{1}{\delta}}{\eta\varepsilon})$ times. To ensure bounded support of the algorithm over *all* possible inputs, we additionally add $\log(1/\delta)/\varepsilon$ copies of some fixed 'dummy hypothesis' $y \in \mathcal{Y}$ to the dataset, so DP selection always outputs something in $M$ by design. While this dummy hypothesis $y$ may not be correct, strong correctness still holds because the only way $y$ ends up in the support is if many non-heavy hitters lie in $M$ as well, a low probability event as long as the initial heavy-hitter subroutine succeeds with high probability. Finally, we remark the entire procedure is user-level private since swapping out a full sample to the heavy-hitter estimation procedure (one user) only changes one element in the database $M$.

In the reverse direction, we are given a $(T, \varepsilon, \delta)$-user-level DP algorithm $\mathcal{A}_{DP}$ on $\ell$ random bits which, by (Ghazi et al., 2024), we may also assume is $(.5, .5, .5)$-perfectly

generalizing. We refer the reader to Definition D.6 for the exact definition, and for now note this implies the existence of a sample $S$ such that for all $\mathcal{O} \subset \mathcal{Y}$ we have $\Pr_r[\mathcal{A}_{DP}(S; \cdot) \in \mathcal{O}] \leq e^{1/2} \Pr_{S', r}[\mathcal{A}_{DP}(\cdot, \cdot) \in \mathcal{O}] + 1/2$. Since $\mathcal{A}_{DP}(S)$ has support size at most $2^\ell$, setting $\mathcal{O} = \text{Supp}(\mathcal{A}_{DP}(S))$ (and therefore the LHS to 1) implies $\Pr_{S', r}[A(\cdot, \cdot) \in \text{Supp}(\mathcal{A}_{DP}(S))] \geq \Omega(1)$, or equivalently, that $\mathcal{A}_{DP}$ has a $O(2^{-\ell})$-heavy-hitter. We may then use the same procedure as in Section 4.1 to transform this into a deterministic $O(2^{-\ell})$-globally stable algorithm as desired.

## 4.3. The Stability of Agnostic Learning

Our agnostic globally stable learner is based on the agnostic-to-realizable reduction framework of (Hopkins et al., 2022), in particular a variant for replicable algorithms in (Bun et al., 2023). In other words, starting with an $\eta$-globally stable learner $\mathcal{A}_{\text{PAC}}$ for the 'realizable setting' (where it is assumed the underlying distribution is labeled by a hypothesis in $H$), we will build a globally stable agnostic learner $\mathcal{A}_{\text{Agn}}$ for arbitrary distributions. Our starting point is therefore the realizable setting, where we rely on the following result of (Ghazi et al., 2021a): every class $(X, H)$ with finite Littlestone dimension $d$ has an $\eta = 2^{\text{poly}(d)} \alpha^{-O(1)}$-globally stable learner on $n(\alpha, \beta) = \text{poly}(d, \alpha, \log(1/\beta))$ samples in the realizable setting.

The core idea behind the reduction is simple: we draw a large unlabeled sample $S_U$ of size $n(\alpha, \beta)$ and run $\mathcal{A}_{\text{PAC}}$ across all possible labelings of $S_U$ in the class, then output a random hypothesis in the resulting set of outputs with low empirical error (tested across a fresh batch of labeled samples from the underlying agnostic distribution). This is an agnostic learner because the first stage always runs $\mathcal{A}_{\text{PAC}}$ over the labeled sample $(S_U, h_{OPT}(S_U))$ for some optimal hypothesis $h_{OPT}$ by design, and therefore with high probability has an output that is close to the best hypothesis in $H$. Moreover, because $\mathcal{A}_{\text{PAC}}$ is globally stable, there is even some *fixed* good hypothesis $h_D$ close to $h_{OPT}$ that appears in the set with probability at least $\eta$, and therefore is given as the final output with probability $\eta / |S_U|^{O(VC(H))}$, where $VC(H)$ is the VC dimension and the divisor $|S_U|^{O(VC(H))}$ is an upper bound on the number of possible labelings of $S_U$ by $H$ (see Lemma A.13).

This procedure almost works, but it has a core issue: to achieve confidence $\beta$, we have to set $|S_U|$ scaling with $\beta$, but then the global stability depends on $\beta$ as well which is not allowed. We fix this by running $\mathcal{A}_{\text{PAC}}$ only with confidence roughly $\eta$, ensuring at least its heavy hitter is an $\alpha$-optimal hypothesis. Unfortunately, this means we are no longer guaranteed the resulting set of outputs will have an $\alpha$-optimal hypothesis with high probability, breaking the argument. We fix this by running the procedure over $T = \frac{\log(1/\beta)}{\eta}$ independent unlabeled samples, which ensures the nearly-optimal

heavy hitter $h_D$ occurs in the set with probability at least $1 - \beta$. Naively we are then back at the same problem, since the output set has size $Tn(\alpha, \eta)^{O(VC)}$, but the key is to realize that $h_D$ not only occurs with high probability, it occurs *at least $\Omega(\eta T)$ times* with high probability. This means we can prune all hypotheses from the resulting list that don't occur $\Omega(\eta T)$ times, cutting the total number down to at most $\frac{1}{\eta} n(\alpha, \eta)^{O(VC(H))}$. This results in an agnostic learner with a $\frac{1}{\eta} n(\alpha, \eta)^{O(VC(H))}$-heavy-hitter, which can be converted into a globally-stable learner by our prior procedure.

As written, the above has exponential sample complexity in the Littlestone dimension – this can be fixed in the heavy-hitter setting by exploiting a stronger guarantee of (Ghazi et al., 2021a)'s algorithm which in reality outputs a list of $\exp(\text{poly}(d)) \alpha^{-O(1)}$ hypotheses such that some $h_D$ appears in the list with probability at least $\Omega(\frac{1}{d})$. We can then perform the same procedure about but set $T = \text{poly}(d, \beta)$ to get the same result with improved sample complexity.

## Acknowledgements

We thank Zachary Chase for many helpful discussions on list stability and differential privacy, especially for referring us to (Ghazi et al., 2021a)'s list-stability bound with polynomial dependence in excess error in the realizable regime, and suggesting how to make list-replicable learners deterministic.

## Impact Statement

This paper presents work whose goal is to advance the field of Machine Learning. There are many potential societal consequences of our work, none which we feel must be specifically highlighted here.

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

# A. Preliminaries

We now cover our notions of stability and general setting in more formality. Most of our results are proved for the following standard and widely applicable notion of a statistical task (see e.g. (Bun et al., 2023)):

**Definition A.1** (Statistical tasks). A statistical task $\mathcal{M}$ consists of:

1. A data domain $\mathcal{X}$

2. An output domain $\mathcal{Y}$

3. For every distribution $D$ over $\mathcal{X}$, a subset $G_D \subset \mathcal{Y}$ of 'accepted solutions'

An **algorithm** solving $\mathcal{M}$ is a (possibly randomized) mapping $\mathcal{A} : \mathcal{X}^* \to \mathcal{Y}$ such that for all $\beta > 0$ (the '**confidence**'), there is a function $n = n(\beta)$ (the '**sample complexity**') such that

$$\forall D : \Pr_{S \sim D^n}[\mathcal{A}(S) \in G_D] \geq 1 - \beta.$$

We remark that in some statistical tasks (e.g. realizable PAC Learning), one only considers a restricted family of distributions $D$ over the data domain. All our results extend immediately to this generalized setting—we have chosen to focus on the distribution-free definition only for simplicity of exposition.

## A.1. The Stability Zoo

We now overview the main notions of stability studied in this work. We remark that our definitions differ slightly from some prior works studying stability in *learning*, which typically comes with an additional accuracy parameter. We discuss this distinction further in Appendix A.4.

We start with the core notion of global stability, which measures the best replication probability achieved by any determinstic algorithm solving the task.

**Definition A.2** (Global stability). Let $\mathcal{M}$ be a statistical task. An algorithm $\mathcal{A}$ solving $\mathcal{M}$ is said to be $\eta$-globally stable if for large enough $n$ and every distribution $D$

$$\Pr_{S,S' \sim D^n}[\mathcal{A}(S) = \mathcal{A}(S')] \geq \eta.$$

The *g-stable complexity* of $\mathcal{M}$, denoted $C_{\text{Glob}}(\mathcal{M})$, is the infimum of $\log(\frac{1}{\eta})$ taken over all deterministic $\eta$-globally stable algorithms solving $\mathcal{M}$. If the set of such algorithms is empty, we write $C_{\text{Glob}} = \infty$.

As mentioned previously, this differs from the 'traditional' notion of global stability in (Bun et al., 2020), which only requires $\mathcal{A}$ to have an $\eta$-heavy-hitter (and may use unshared internal randomness), a seemingly weaker requirement. We show these notions are fully equivalent in Appendix B.

Global stability can never achieve replication probability beyond $1/2$ for any non-trivial statistical task (Chase et al., 2023). Motivated by this fact, (Impagliazzo et al., 2022) introduced *replicability*, allowing the use of shared internal randomness to boost replication success beyond the $\frac{1}{2}$ barrier.

**Definition A.3** (Replicability). Let $\mathcal{M}$ be a statistical task. An algorithm $\mathcal{A}$ solving $\mathcal{M}$ is said to be $\rho$-replicable if for all distributions $D$:

$$\Pr_{r,S,S' \sim D^n}[\mathcal{A}(S; r) = \mathcal{A}(S'; r)] \geq 1 - \rho.$$

In (Dixon et al., 2023), the authors introduce *certificate complexity* measuring the least number of shared random bits required to achieve $\rho$-replicability. Since we are interested in comparing certificate complexity to global stability, we will instead work mostly with a natural 'parameter-free' version of this definition. In particular, we will study the number of random strings required to achieve replicability strictly better than $\frac{1}{2}$. This is the natural threshold at which many random strings are forced to have a unique 'canonical' hypothesis, and can be easily amplified to arbitrary $\rho$.

**Definition A.4** (Certificate Complexity). The **certificate complexity** of a statistical task $\mathcal{M}$, denoted $C_{\text{Rep}}(\mathcal{M})$, is the smallest number of shared random bits over which there exists a $> \frac{1}{2}$-replicable algorithm solving $\mathcal{M}$, i.e. some $\mathcal{A}$ satisfying:

$$\Pr_{r \sim \{0,1\}^{C_{\text{Rep}}}, S, S' \sim D^n} [\mathcal{A}(S; r) = \mathcal{A}(S'; r)] > 1/2$$

If no such algorithm exists for any finite $C_{\text{Rep}}$, we write $C_{\text{Rep}} = \infty$.

We remark that once one has achieved $> \frac{1}{2}$ replicability, Theorem 2.1 implies an amplification procedure to achieve any $\rho$-replicability parameter using an additional $\log \frac{1}{\rho}$ random bits.

For the rest of the work, we usually omit $\mathcal{M}$ and just write $C_{\text{Glob}}$ and $C_{\text{Rep}}$ when clear from context.

## A.2. Differential Privacy

Global stability (Bun et al., 2020) and replicability (Ghazi et al., 2021b) (known as 'pseudo-global stability' in this context) were both introduced as a tool to study the widely influential notion of *differential privacy*, a powerful algorithmic guarantee promising an algorithm has similar output distributions over neighboring samples.

**Definition A.5** (Differential Privacy ((Dwork et al., 2006b;a))). Let $\mathcal{M}$ be a statistical task, and $\varepsilon, \delta > 0$. Two samples $S, S' \in (\mathcal{X}^n)$ are said to be neighboring if they differ in exactly one coordinate. An algorithm $\mathcal{A}$ solving $\mathcal{M}$ is said to be $(\varepsilon, \delta)$-differentially private if for any neighboring samples $S, S'$ and measurable events $\mathcal{O} \subset \mathcal{Y}$:

$$\Pr[\mathcal{A}(S) \in \mathcal{O}] \leq e^{\varepsilon} \Pr[\mathcal{A}(S') \in \mathcal{O}] + \delta$$

We will sometimes use the notation $\mathcal{A}(S) \stackrel{(\varepsilon, \delta)}{=} \mathcal{A}(S')$ to denote the above closeness guarantee.

In early works (Bun et al., 2020; Ghazi et al., 2021a;b; 2024; Bun et al., 2023), global stability and its variants played a key role in bounding the sample complexity of differentially private PAC learning. Quite recently, (Canonne et al., 2024) expanded this connection by using techniques developed for certificate complexity (Dixon et al., 2023; Woude et al., 2023) to bound the number of random bits needed to achieve differential privacy, a parameter we'll call *DP complexity*.

**Definition A.6** (Parametrized DP Complexity). The $(n, \beta, \varepsilon, \delta)$-DP Complexity of a statistical task $\mathcal{M}$ is the smallest integer $C_{DP}(n, \beta, \varepsilon, \delta)$ such that there exists an $(\varepsilon, \delta)$-DP algorithm on $n$ samples solving $\mathcal{M}$ with confidence $\beta$ using only $C_{DP}(n, \beta, \varepsilon, \delta)$ random bits. If no such algorithm exists for any finite $C_{DP}(n, \beta, \varepsilon, \delta)$, we write $C_{DP}(n, \beta, \varepsilon, \delta) = \infty$.

Unlike certificate complexity, it is not clear there is a natural 'parameter-free' variant of DP complexity, especially since the concept frequently is used at many different scales of $(\varepsilon, \delta)$ in theory and practice, themselves dependent on $n$ and $\beta$. As such, we will focus mostly on parameter-dependent translations in this setting, and define the following 'parametrized' variants of g-stable and certificate complexity.

**Definition A.7** (Parametrized g-Stable Complexity). The $(n, \beta)$-g-stable complexity of a task $\mathcal{M}$, denoted $C_{\text{Glob}}(n, \beta)$, is $\log \frac{1}{\eta}$ where $\eta$ is the largest global stability achieved by any $\beta$-confident algorithm on $n$ samples solving $\mathcal{M}$. If no such algorithm exists, we write $C_{\text{Glob}}(n, \beta) = \infty$.

**Definition A.8** (Parametrized Certificate Complexity). The $(n, \beta)$-certificate complexity of a task $\mathcal{M}$, denoted $C_{\text{Rep}}(n, \beta)$, is the smallest number of random bits such that there is a $\beta$-confident, better than $\frac{1}{2}$-replicable algorithm on $n$ samples solving $\mathcal{M}$. If no such algorithm exists, we write $C_{\text{Rep}}(n, \beta) = \infty$.

## A.3. User Level Privacy

In (Ghazi et al., 2021b; Bun et al., 2023), the authors take advantage of the strong stability guarantees of replicability to build '*user-level*' private algorithms (Levy et al., 2021), a stronger notion of privacy in which each user is thought of as providing many data points and the goal is to protect against an adversary swapping out a single user's *entire* dataset. Formally, the input dataset $S$ is actually split into subsets $S_1, \ldots, S_T$ coming from $T$-separate users, and we bound the algorithms deviation upon switching out one such $S_i$:

**Definition A.9** (User-Level Differential Privacy). Let $\mathcal{M}$ be a statistical task, $n, T \in \mathbb{N}$ such that $T$ divides $n$, and call samples $(S_1, \ldots, S_T), (S'_1, \ldots, S'_T) \in (\mathcal{X}^{n/T})^T$ neighboring if $S'_j = S_j$ for all but one $j \in [T]$. We call an algorithm $\mathcal{A}$ solving $\mathcal{M}$ $T$-user $(\varepsilon, \delta)$-DP if $\mathcal{A}(S) \stackrel{(\varepsilon, \delta)}{=} \mathcal{A}(S')$ for all such neighboring data-sets $S, S'$

User-level DP allows us to tighten the connection between DP complexity and certificate complexity, giving a tight transformation between the two up to constants for reasonable regimes of $\varepsilon$ and $\delta$.

**Definition A.10** (User-Level DP Complexity). The $(T, \varepsilon, \delta)$-user-level-DP Complexity of a statistical task $\mathcal{M}$ is the smallest integer $C_{DP}(T, \varepsilon, \delta)$ such that there exists a $T$-user $(\varepsilon, \delta)$-DP algorithm solving $\mathcal{M}$ using only $C_{DP}(T, \varepsilon, \delta)$ random bits. If no such algorithm exists for any finite $C_{DP}(T, \varepsilon, \delta)$, we write $C_{DP}(T, \varepsilon, \delta) = \infty$.

## A.4. PAC Learning

Having established close quantitative connections between globally stable, certificate, and DP complexity, our second focus is to concretely determine the randomness complexity of a fundamental statistical task: binary classification. We will focus on the standard PAC framework, in which we have a data domain $X$ and a *hypothesis class* $H = \{h : X \to \{0,1\}\}$, a collection of possible labelings of $\mathcal{X}$. Let $D$ be a distribution over labeled samples ($X \times \{0,1\}$) and $h : X \to \{0,1\}$ a potential labeling. The error of $h$ under $D$ is

$$err_D(h) = \Pr_{(x,y) \sim D}[h(x) \neq y].$$

Similarly the empirical error of $h$ on a labeled sample $S$ is

$$err_S(h) = \Pr_{(x,y) \in S}[h(x) \neq y].$$

We write the best possible error achieved over $H$ as

$$OPT_D := \inf_{h \in H} err_D(h).$$

We call a hypothesis $h$ $\alpha$-optimal if it achieves error at most $OPT + \alpha$. An algorithm is said to (agnostic) PAC learn the class $(\mathcal{X}, H)$ if for every $\alpha, \beta > 0$, on sufficiently many samples $\mathcal{A}$ outputs an $\alpha$-optimal hypothesis with probability at least $1 - \beta$.

**Definition A.11** ((Agnostic) PAC Learning). We say a class $(X, H)$ is PAC-Learnable if $\forall \alpha, \beta > 0$ there exists $n = n(\alpha, \beta)$ and a (possibly randomized) algorithm $\mathcal{A} : \mathcal{X}^n \to P(\mathcal{X})$ satisfying

$$\Pr_{S \sim D^n}[err_D(\mathcal{A}(S)) > OPT_D + \alpha] < \beta.$$

Note that PAC-Learning, as formalized above, corresponds to an *infinite sequence* of statistical tasks $\{\mathcal{M}^{(X,H)}(\alpha)\}_{\alpha > 0}$ parametrized by the accuracy $\alpha$, namely where the data domain is $\mathcal{X} = X \times \{0,1\}$, the output space is all possible labelings $\mathcal{Y} = \{h : X \to \{0,1\}\}$, and the set of accepted solutions is $G_D := \{h \in H : err_D(H) \leq OPT + \alpha\}$. With this in mind, we may therefore define the 'error-dependent' stable/certificate/DP-complexity of a class $(X, H)$ as

$$C_{\text{Glob}}(\alpha) = C_{\text{Glob}}(\mathcal{M}(\alpha)),$$

and likewise for certificate and DP complexity. We emphasize this differs from prior works (Bun et al., 2020; Chase et al., 2024) that focus on g-stable complexity bounds that are *independent* of the excess error $\alpha$ (a distinction we will discuss further shortly).

The sample complexity of PAC learning (with no stability constraints) is tightly controlled by a combinatorial parameter called the VC-dimension:

**Definition A.12** (VC Dimension). We say a subset $S = \{x_1, \ldots, x_d\}$ is *shattered* by $(X, H)$ if all possible $2^d$ labelings of $S$ can be achieved by $H$. The VC dimension of $(X, H)$ is the size of the largest shattered set.

We will need the following classical lemma bounding the number of labelings of any subset of data as a function of the VC dimension.

**Lemma A.13** (The Sauer-Shelah-Perles Lemma (Vapnik & Chervonenkis, 1974; Sauer, 1972; Shelah, 1972)). *Let $(X, H)$ be a hypothesis class. For any $n \in \mathbb{N}$ and $S \subset X$ of size $n$, the number of labelings of $S$ by hypotheses in $H$ is at most $n^{O(VC(H))}$.*

Unlike standard learning, learning under replicability or DP constraints is not possible for every VC class. Instead, these notions are characterized by a stronger definition arising from online learning called the *Littlestone dimension* (Bun et al., 2020; 2023). The notion is based on *mistake trees*, complete binary trees in which every internal node is labeled by an element $x \in X$. Every root to leaf path in a depth $d + 1$ mistake tree then corresponds to a sequence $(x_1, y_1), \ldots, (x_d, y_d)$, where $y_i \in \{0, 1\}$ indicates whether the right or left path is taken down the tree. A hypothesis class $(X, H)$ *shatters* the tree if for every such path there exists a hypothesis $h \in H$ such that $h(x_i) = y_i$ for all $i \leq d$. The Littlestone dimension is the largest depth of any shattered mistake tree:

**Definition A.14** (Littlestone Dimension). The Littlestone dimension of a class $(X, H)$ is the largest depth of any mistake tree shattered by $H$.

We note that the Littlestone dimension is always at least as large as the VC dimension.

In (Bun et al., 2020), the authors prove that any class with finite Littlestone dimension $d$ has a universal upper bound on the g-stable complexity of $2^{O(d)}$ under the assumption that $\text{OPT} = 0$. (Chase et al., 2024) showed this result cannot extend in the full agnostic setting, and indeed that *no* error-independent bound is possible for any infinite class. They ask whether it is possible to give an error-dependent bound, a question we resolve in the positive in Theorem 2.4.

## B. Global Stability: Replication vs Heavy-Hitters

In this section we prove that Definition A.2 (replication global stability) is equivalent to the traditional randomized 'heavy-hitters' variant studied in (Bun et al., 2020; Chase et al., 2023). Given a distribution $D$ over some domain $X$, an element $x \in X$ is called an $\eta$-heavy-hitter if the measure of $x$ in $D$ is at least $\eta$. Traditional global stability promises the existence of a heavy hitter over the output distribution of the algorithm.

**Definition B.1** (Heavy-Hitter Global stability (Bun et al., 2020)). Let $\mathcal{M}$ be a statistical task. An algorithm $\mathcal{A}$ solving $\mathcal{M}$ is said to be $\eta$-heavy-hitter globally stable if for large enough $n$ and every distribution $D$ there exists $h_D$ such that:

$$\Pr_{S, S' \sim D^n}[\mathcal{A}(S) = h_D] \geq \eta.$$

The *HH-stable complexity* of $\mathcal{M}$, denoted $C_{\text{Glob-HH}}$ is the infimum of $\log(\frac{1}{\eta})$ taken over all (possibly randomized) $\eta$-heavy-hitter globally stable algorithms solving $\mathcal{M}$.

In (Chase et al., 2023), the authors observe that HH-global stability and global stability as in Definition A.2 differ by at most a quadratic factor. We prove they are actually the same quantity, a fact we will use later both to move between randomized and deterministic algorithms, and to move from heavy-hitter bounds to global stability.

**Theorem B.2** (Global Stability vs HH-Global Stability). *For any statistical task $\mathcal{M}$: $C_{\text{Glob-HH}} = C_{\text{Glob}}$*

*Proof.* We remark that any $\eta$-globally stable algorithm is automatically $\eta$-heavy-hitter globally stable, so $C_{\text{Glob-HH}} \geq C_{\text{Glob}}$ is immediate. Thus it remains to prove $C_{\text{Glob}} \geq C_{\text{Glob-HH}}$.

Write $\eta = 2^{-C_{\text{Glob-HH}}}$. For any $\gamma, \beta > 0$, we are promised a $\beta$-correct, possibly randomized algorithm $\mathcal{A}$ such that for any distribution $D$ there exists a hypothesis $h_D$ satisfying

$$\Pr_{r, S \sim D^n}[\mathcal{A}(S) = h_D] \geq \eta - \gamma$$

Naively, $\mathcal{A}$ only has replication probability approaching $\eta^2$. To improve this, the idea is to build a new algorithm that outputs one of $\mathcal{A}$'s $(\eta - 2\gamma)$-heavy-hitters with probability at least $1 - \gamma$. By standard concentration bounds (see e.g. (Kontorovich & Painsky, 2024)) this can be done simply by running $\mathcal{A}$ on $O(\frac{\log(1/\gamma)}{\eta^2})$ fresh samples and outputting the most common hypothesis in the list (breaking ties arbitrarily). Toward this end, denote the list of $(\eta - 2\gamma)$-heavy-hitters by $L_D$, and observe that for small enough $\gamma \leq O(\eta)$, $|L_D| \leq \frac{1}{\eta}$.

Taking $\beta \leq O(\eta)$ sufficiently small, we can also ensure all elements of $L_D$ are correct solutions, i.e. that $L_D \subset G_D$. We have therefore arrived at an algorithm $\mathcal{A}_{\text{List}}$ with the following property: for any distribution $D$, there exists a list $L_D \subset G_D$ of size at most $\frac{1}{\eta}$ such that

$$\Pr_{r, S \sim D^n}[\mathcal{A}_{\text{List}}(S; r) \in L_D] \geq 1 - \gamma$$

We note algorithms satisfying this guarantee are called '$|L_D|$-list-replicable' (Chase et al., 2023; Dixon et al., 2023).

We are now almost done since conditioned on landing in $L_D$ in both runs, the replication probability of $\mathcal{A}_{\text{List}}$ is at least $\frac{1}{|L_D|} \geq \eta$, and the probability of being outside the list can be taken to arbitrarily small. The only issue is that $\mathcal{A}_{\text{List}}$ is randomized. Thus it is sufficient to argue we can de-randomize $\mathcal{A}_{\text{List}}$ at the cost of increasing the list-failure probability $\gamma$ to some corresponding $\gamma'$ also arbitrarily small.

We do this simply by passing to the most likely output of $\mathcal{A}_{\text{List}}$. Namely define $\mathcal{A}_{\det}$ by

$$\mathcal{A}_{det}(S) := \underset{y \in \mathcal{Y}}{\operatorname{argmax}}\{\Pr[\mathcal{A}_{\text{List}}(S) = y]\},$$

and note that $\mathcal{A}_{det}(S)$ can be easily computed by simply running $\mathcal{A}_{\text{List}}(S; r)$ for all choices of internal randomness $r$ and outputting the most common result (breaking ties arbitrarily). We claim:

$$\Pr_S[\mathcal{A}_{det}(S) \in L_D] = \Pr_S\left[\underset{y \in Y}{\operatorname{argmax}}\{\Pr[\mathcal{A}_{\text{List}}(S) = y]\} \in L_D\right] \geq 1 - \frac{2\gamma}{\eta}.$$

This follows from the observation that for any sample $S$ whose maximal output is not in $L_D$, the probability of outputting an element outside of $L_D$ is then at least $\frac{\eta}{2}$ (either the maximum probability, by assumption achieved by an element outside of $L_D$ is at least $\frac{\eta}{2}$ in which case we are done, or is at most $\frac{\eta}{2}$ in which case the list elements make up only $1/2$ the mass by assumption). Finally since we may take $\gamma$ arbitrarily small with respect to $\eta$, taking a large enough sample we may make $\mathcal{A}_{\det}$ $\eta - \gamma'$ globally stable for any $\gamma' > 0$, implying $C_{\text{Glob}} \geq \log \frac{1}{\eta} = C_{\text{Glob-HH}}$ as desired. □

It will also be useful to have a parametrized version of the above transform:

**Corollary B.3.** *For any statistical task $\mathcal{M}$ and $\eta, \beta > 0$, given an $\eta$-HH globally stable, $\beta$-confident algorithm for $\mathcal{M}$ on $n$ samples, the transform in Theorem B.2 gives an $O(\eta)$-globally stable $\beta'$-confident algorithm for $\mathcal{M}$ on $n'$ samples, where $\beta' \leq O(\beta/\eta)$ and $n' \leq n \cdot \tilde{O}\left(\frac{\log\frac{1}{\beta}}{\eta^2}\right)$.*

*Proof.* Note that the statement is trivial if $\beta > \eta$, and take $\gamma, \gamma' \leq O(\beta\eta)$ in the proof of Theorem B.2. For such parameters $\mathcal{A}_{\det}$ runs $\mathcal{A}$ at most $\tilde{O}\left(\frac{\log\frac{1}{\beta}}{\eta^2}\right)$ times, and has confidence $\beta' \leq 2\gamma/\eta \leq \beta/\eta$ (since it outputs an element in $L_D$ with at least this probability, which are correct by the assumption on $\beta$). □

## C. Global Stability and Certificate Complexity

In this section, we prove Theorem 2.1, the (near) equivalence of g-stable and certificate complexity. We repeat the main result of the section here for convenience.

**Theorem C.1** (Global Stability vs Certificate Complexity). *For any statistical task $\mathcal{M}$:*

$$C_{Glob} \leq C_{Rep} \leq C_{Glob} + 1.$$

*Moreover, the number of random bits required to achieve $\rho$-replicability is at most $\lceil C_{Glob} + \log(1/\rho) \rceil$*

We start with the easy direction: certificate complexity to global stability. It is easy to see that any better-than-half replicable algorithm on $\ell$-bits is automatically $2^{-\ell-1}$-globally stable (albeit randomized), since there must exist some $r \in \{0,1\}^\ell$ which has $> \frac{1}{2}$ replication probability. This can be amplified to probability $2^{-\ell} - \eta$ for any $\eta > 0$ simply by running the algorithm many times and taking majority, then made deterministic by Theorem B.2.

*Proof of Theorem C.1 (Lower Bound).* Recall by Theorem B.2, it is enough to prove for any $\tau > 0$ the existence of a *randomized* algorithm $\mathcal{A}$ solving $\mathcal{M}$ such that for any distribution $D$

$$\Pr_{S,S' \sim D, r, r'}[\mathcal{A}(S; r) = \mathcal{A}(S'; r')] \geq 2^{-C_{\text{Rep}}} - \tau$$

where $r$ and $r'$ are independent random strings.

Now by assumption there exists a $(\frac{1}{2} + \gamma)$-replicable algorithm solving $\mathcal{M}$ using $C_{\text{Rep}}$ random bits. By averaging, we then have that for any distribution $D$, there exists a string $r_D \in \{0,1\}^{C_{\text{Rep}}}$ such that

$$\Pr_{S,S'\sim D^n}[\mathcal{A}(S;r) = \mathcal{A}(S';r)] \geq \frac{1}{2} + \gamma$$

and therefore that there exists a 'canonical hypothesis' $h_D$ such that

$$\Pr_{S,S'\sim D^n}[\mathcal{A}(S;r) = h_D] \geq \frac{1}{2} + \gamma$$

Given this, consider the 'majority amplified' algorithm $\mathcal{A}_{\text{maj-}T}$ on $nT$ samples and the same internal randomness defined as

$$\mathcal{A}_{\text{maj-}T}(S_1, \ldots, S_T; r) = \text{plurality}\{\mathcal{A}(S_i; r)\}$$

breaking ties arbitrarily. Taking $T = O\left(\frac{\log \frac{1}{\tau}}{\gamma^2}\right)$ large enough, Chernoff promises

$$\Pr_{S\sim D^{nT}}[\mathcal{A}_{\text{maj-}T}(S;r) = h_D] \geq 1 - \frac{\tau}{2},$$

so the total collision probability over two runs on independent strings is at least

$$\Pr[\mathcal{A}(S;r) = \mathcal{A}(S';r')] \geq \Pr[r = r' = r_D] \cdot \Pr[\mathcal{A}(S;r_D) = \mathcal{A}(S';r_D) \mid r = r' = r_D]$$
$$\geq 2^{-C_{\text{Rep}}}(1-\tau)$$

as desired. □

We now prove the reverse direction, which follows from a careful discretization of (Impagliazzo et al., 2022)'s random thresholding method. We first give the algorithm in pseudocode.

---

**Algorithm 1** Global Stability $\to$ Certificate Complexity

---

**Result:** $(\rho - \tau')$-replicable, $\beta$-confident algorithm on $\lceil \frac{2^{C_{\text{Glob}}} - 1}{\rho} \rceil$ random strings

**Input:** $(2^{-C_{\text{Glob}}} - \tau)$-globally stable and $O(2^{-C_{\text{Glob}}})$-confident algorithm $\mathcal{A}$, total ordering $\phi$ of $\mathcal{Y}$.

**Parameters:**

- Threshold number $T = \lceil \frac{2^{C_{\text{Glob}}} - 1}{\rho} \rceil$

- Heavy-Hitter parameter $\eta = 2^{-C_{\text{Glob}}}$

- Threshold offset $\tau \ll \gamma \ll \eta$

- Amplification parameter $N(\gamma, \beta, \tau')$

**Algorithm:**

1. Run $\mathcal{A}$ across $N$ fresh samples $S \sim D^n$ and let $\hat{p}_h$ denote the empirical density of each $h \in \mathcal{Y}$

2. Select $i \in [T]$ uniformly at random, and let

$$H_i := \{h : \hat{p}_h \geq \eta - i\gamma\}$$

**Return** $\underset{H_i}{\arg\min} \, \hat{p}_h$ (breaking ties via $\phi$) or $\perp$ if $H_i = \varnothing$

---

*Proof of Theorem C.1 (Upper Bound).* We prove a slightly stronger result: for any $\tau' > 0$, there is a $(\rho - \tau')$-replicable learner using $T = \lceil \frac{2^{C_{\text{Glob}}} - 1}{\rho} \rceil$ random strings. Achieving $\rho > \frac{1}{2} + \frac{1}{2^{O(C_{\text{Glob}})}}$ can therefore be done in $C_{\text{Glob}} + 1$ random bits,

and general $\rho$ may be achieved using $\lceil C_{\text{Glob}} + \log(1/\rho) \rceil$ bits as desired.[7]

To prove the stronger statement, first observe that by standard concentration inequalities (see e.g. (Kontorovich & Painsky, 2024)), for large enough $T_1 \leq O(\frac{\log(\frac{1}{\beta\tau'})}{\gamma^2})$ with probability at least $1 - \beta\tau'$ all empirical estimates $\hat{p}_h$ in Step (1) of Algorithm 1 satisfy

$$|\hat{p}_h - p| < \gamma/3. \tag{2}$$

Condition on Equation (2) occurring. Choosing $\tau \leq O(\gamma) \leq O(T^{-1})$ sufficiently small, there are at most $\frac{1}{\eta}$ hypotheses of empirical weight greater than $\eta - T\gamma$. Since $\mathcal{A}$ must have an $(\eta - \tau)$-heavy hitter, there also exists at least one $\hat{p} \geq \eta - \tau - \gamma/3 > \eta - 2\gamma/3$, so the set of empirical heavy hitters $H_i$ (defined in Algorithm 1 Step 2) is non-empty for any $i$ and Algorithm 1 will not output $\perp$.

Now consider the set of $T$ thresholds $\{\eta - \gamma, \eta - 2\gamma, \ldots, \eta - T\gamma\}$. By assumption, at most $\frac{1}{\eta} - 1$ of these thresholds have a hypothesis with true weight within $\gamma/3$ of their value. Thus choosing a random threshold, the probability we select one with no nearby hypothesis is at least $1 - \frac{\frac{1}{\eta} - 1}{T}$. Conditioned on selecting such a threshold $t$ and Equation (2), the set of hypotheses with empirical measure greater than $t$ is always the same, so the algorithm always outputs the smallest such element in the list according to $\phi$ as desired. In total, it follows the algorithm is $\left( \frac{2^{C_{\text{Glob}}} - 1}{\lceil \rho^{-1}(2^{C_{\text{Glob}}} - 1) \rceil} - \tau' \right)$-replicable

Finally, since the algorithm is assumed to be $O(2^{-C_{\text{Glob}}})$-confident, any true $\Omega(\eta)$-heavy hitter must also be correct. Conditioned on Equation (2) Algorithm 1 always outputs such a heavy hitter, so the final algorithm is $\beta$-confident as well. $\qquad\square$

We note that in the setting of achieving strictly greater than $\frac{1}{2}$ replicability, the sample overhead in Theorem C.1 can be taken as at worst $2^{O(C_{\text{Glob}})}$, since all parameters $\tau, \tau', \gamma$ can be set to $2^{-O(C_{\text{Glob}})}$ and still achieve the desired replicability. We will use this fact in proving Theorem 2.4.

## D. Stability and Differential Privacy

We now prove Theorem 2.2 and Theorem 2.3 connecting global stability and the randomness complexity of differential privacy. We restate the results in more formality.

**Theorem D.1** (Stability Boosting (DP)). *There exists a universal constant $c > 0$ such that for any statistical task $\mathcal{M}$:*

*1. (Stability to DP):* $C_{DP}(n, \beta, \varepsilon, \delta) \leq C_{Glob}(n', \beta') + \log(1/\varepsilon) + \log(1/\delta) + \log\log(1/\delta) + O(1)$

*for any $n \geq n' \cdot O(\frac{2^{3C_{Glob}(n,\beta)} \log \frac{1}{\beta} \log \frac{1}{\delta}}{\varepsilon})$, $\beta' \geq 2^{C_{Glob}(n,\beta)} \frac{\log \frac{1}{\delta}}{c\varepsilon} (\beta 2^{C_{Glob}(n,\beta)+1})^{O(\frac{\log \frac{1}{\delta}}{\varepsilon})}$, and*

*2. (DP to Stability):* $C_{Glob}(n, \beta) \leq C_{DP}(n', \beta', \varepsilon, \delta) + O(1)$

*for any $\varepsilon \leq \frac{c}{\sqrt{n' \log(n')}}$ and $\delta \leq \frac{c}{n'}$, $n \geq n' \cdot \tilde{O}\left( 2^{2C_{DP}(n,\beta,\varepsilon,\delta)} \log \frac{1}{\beta} \right)$, and $\beta' \geq O(\beta 2^{C_{DP}(n,\beta,\varepsilon,\delta)})$.*

**Theorem D.2** (Stability Boosting (User-Level DP)). *There exist universal constants $c_1, c_2 > 0$ such that for any statistical task $\mathcal{M}$:*

*1. (Stability to DP):* $C_{DP}\left( 2^{C_{Glob}} \frac{c_1 \log \frac{1}{\delta}}{\varepsilon}, \varepsilon, \delta \right) \leq C_{Glob} + \log(1/\varepsilon) + \log(1/\delta) + \log\log(1/\delta) + O(1)$

*2. (DP to Stability):* $C_{Glob} \leq C_{DP}(T, \varepsilon, \delta) + O(1)$

*where the latter holds for any $(T, \varepsilon, \delta)$ satisfying $\varepsilon \leq \frac{c_2}{\sqrt{T \log(T)}}$ and $\delta \leq \frac{c_2}{T}$.*

---

[7]Formally, we note there is some discrepancy here in sampling a random threshold for $T$ not a power of 2 when $r$ is a bit-string. However, in the argument below increasing the number of random strings/thresholds to the nearest power of 2 only helps replicability, so this is not an actual issue.

Both Theorem D.1 and Theorem D.2 follow as corollaries of the same underlying stability to DP and DP to stability transformations. We start by proving the forward direction: a randomness efficient stability-to-DP transform. The key to achieving low randomness is the following useful observation of (Canonne et al., 2024): any distribution with small support can be approximately sampled using few random bits.

**Lemma D.3** ((Canonne et al., 2024), Lemma 2.10 (rephrased)). *For any randomized algorithm $M : X^n \rightarrow Y$ and $\eta > 0$, there exists an algorithm $M'$ using $\max_x \log(|Supp(M(x))|) + \log(1/\eta)$ random bits such that for every input $x$, $M'$ satisfies:*

1. *(Closeness): $d_{TV}(M(x), M'(x)) \leq \eta$*

2. *(Subset Support): $Supp(M'(x)) \subseteq Supp(M(x))$*

The result will now follow if we can prove a stability-to-DP transformation with two key properties

1. For every $S$, the support of $\mathcal{A}(S)$ is small

2. With high probability over $S$, $\text{Supp}(\mathcal{A}(S)) \subset G_D$,

where we recall $G_D$ is the set of correct solutions for the task $\mathcal{M}$. The stronger correctness property is needed since applying Lemma D.3 might otherwise ruin the correctness of our algorithm.

To build such a transformation, we will use the following DP Selection algorithm also used as the main subroutine in the standard stability-to-DP transformation of prior works (Bun et al., 2020; Ghazi et al., 2021b; Bun et al., 2023).

**Theorem D.4** (DP Selection (Korolova et al., 2009; Bun et al., 2016; 2018), as stated in (Bun et al., 2023)). *There exists some $c > 0$ such that for every $\varepsilon, \delta > 0$ and $m \in \mathbb{N}$, there is an $(\varepsilon, \delta)$-DP algorithm that on input $S \in \mathcal{X}^m$, outputs with probability $1$ an element $x \in X$ that occurs in $S$ at most $\frac{c \log \delta^{-1}}{\varepsilon}$ fewer times than the mode of $S$.*

We can now state and prove our bounded support, strong correctness stability-to-DP transform:

**Lemma D.5.** *Let $\mathcal{M}$ be a statistical task, $\eta, \varepsilon, \delta > 0$, and $T = O\left(\frac{\log(1/\delta)}{\eta\varepsilon}\right)$. Given an $\eta$-globally stable algorithm $\mathcal{A}$ for $\mathcal{M}$ on $n = n(\beta)$ samples, there exists a $T$-user $(\varepsilon, \delta)$-DP algorithm $\mathcal{A}_{DP}$ on $n' = n(\varepsilon, \delta, \beta)$ samples satisfying:*

1. ***Bounded Support:*** *$\forall S : |Supp(\mathcal{A}_{DP}(S;))| \leq T$*

2. ***Strong Correctness:*** *$\Pr_{S \sim D^n}[Supp(\mathcal{A}_{DP}(S;)) \subset G_D] > 1 - \beta'$*

*for $\beta' \leq T(\frac{2\beta}{\eta})^{O(\frac{\log \frac{1}{\delta}}{\varepsilon})}$ and $n' \leq n(\beta) \cdot O\left(\frac{\log(1/\beta)\log(1/\delta)}{\eta^3\varepsilon}\right)$*

*Proof.* By assumption we are given access to a $\beta$-confident algorithm $\mathcal{A}$ on $n = n(\beta)$ samples that has an $\eta$-heavy hitter $h^*$. In order to ensure strong correctness, we will first need to transform $\mathcal{A}$ into a so-called 'list-replicable' algorithm $\mathcal{A}_{\text{List}}$, that is one such that there exists a small list $L \subset G_D$ of correct hypotheses such that for a large enough $m$:

$$\Pr_{S \sim D^m}[\mathcal{A}_{\text{List}}(S) \in L] \geq 1 - \beta.$$

Assume $\beta \leq \eta/2$ (else the Lemma statement is trivial as $\beta' > 1$). Then every $\eta/2$-heavy-hitter of $\mathcal{A}$ must lie in $G_D$. We build $\mathcal{A}_{\text{List}}$ by simply running $\mathcal{A}$ on $O\left(\frac{\log(\frac{1}{\beta})}{\eta^2}\right)$ fresh samples and outputting the most common result. By standard concentration (Kontorovich & Painsky, 2024), the probability that the output of this procedure is not an $\eta/2$-heavy hitter of $\mathcal{A}$ (of which there are at most $\frac{2}{\eta}$) is at most $\beta$ as desired, and the algorithm uses at most $m = O(n\frac{\log(1/\beta)}{\eta^2})$ samples.

We will now use $\mathcal{A}_{\text{List}}$ to generate a dataset of output hypotheses on which to apply DP selection. More formally, fix some arbitrary 'dummy' output $y \in \mathcal{Y}$ (we use this to ensure the end support is bounded) and consider the following procedure generating a dataset to run DP-Selection:

1. Draw $T = O\left(\frac{\log(1/\delta)}{\eta\varepsilon}\right)$ independent size-$m$ samples from $D$

2. Add the output of $\mathcal{A}_{\text{List}}$ on every sample in $T$ to the dataset

3. Add $\frac{c \log \frac{1}{\delta}}{\varepsilon}$ copies of $y$ for $c$ as in Theorem D.4.

Finally, define $\mathcal{A}_{DP}(S)$ to be the algorithm that outputs the result of DP-Selection on the above dataset.

We first analyze correctness. Note it is enough to argue that with probability at least $1 - \beta'$, the support is contained in $L$, since $L \subset G_D$. By Theorem D.4, the only way the support of $\mathcal{A}_{DP}(S)$ contains a hypothesis outside $L$ is if $\mathcal{A}_{\text{List}}$ outputs $h \notin L$ in Step 2 at least $O(\log(1/\delta)/\varepsilon)$ times. Since the samples are independent, the probability of this occurring is at most

$$\sum_{j=O(\log(1/\delta)/\varepsilon)}^{T} \binom{T}{j} \beta^j \leq T \cdot \left(\frac{2\beta}{\eta}\right)^{O(\log(1/\delta)/\varepsilon)},$$

as desired.

It is left to bound the (user-level) differential privacy and support. It is clear the algorithm is $T$-user $(\varepsilon, \delta)$-DP by construction, taking each user's data to be a full sample given to $\mathcal{A}_{\text{List}}$ in Step 2. Furthermore, because the generated dataset always has a hypothesis appearing more than $\frac{c \log \frac{1}{\delta}}{\varepsilon}$ times by construction, Theorem D.4 promises the output lies in the constructed dataset[8] which has size at most $T + 1$ as desired. $\qquad\square$

The forward direction of both Theorem D.1 and Theorem D.2 are now essentially immediate:

*Proof of Theorem D.1 and Theorem D.2 (Item 1).* Write $C = C_{Glob}(n, \beta)$ for notational simplicity, and let $\mathcal{A}$ be the promised $2^{-C+1}$-globally stable algorithm. By Lemma D.5, we can convert $\mathcal{A}$ into a $T$-user $(\varepsilon, \delta/2)$-DP algorithm $\mathcal{A}_{DP}$ on $n$ samples with $T \leq O\left(2^{C_{Glob}} \cdot \frac{\log(1/\delta)}{\varepsilon}\right)$ such that for all $S$, $|\text{Supp}(\mathcal{A}(S; \cdot))| \leq T$, and with probability at least $1 - \beta$, $\text{Supp}(\mathcal{A}_{DP}(S; \cdot)) \subset G_D$. Applying Lemma D.3 with $\eta = \delta/2$ then gives the desired result, where correctness is maintained since the output of the algorithm on any sample $S$ where $\text{Supp}(\mathcal{A}_{DP}(S; \cdot)) \subset G_D$ remains entirely inside $G_D$ by the subset support property. $\qquad\square$

We now move on to the reverse direction of both results, which follows from the equivalence of approximate differential privacy with another strong notion of stability known as *perfect generalization* (Cummings et al., 2016; Bassily & Freund, 2016; Bun et al., 2023; Ghazi et al., 2024).

**Definition D.6** (Perfect Generalization ((Cummings et al., 2016; Bassily & Freund, 2016)))**.** Fix $\beta, \varepsilon, \delta > 0$. An algorithm $\mathcal{A} : \mathcal{X}^n \to \mathcal{Y}$ is called $(\beta, \varepsilon, \delta)$-perfectly-generalizing if for any distribution $D$ over $\mathcal{X}$, there exists a 'canonical distribution' $\text{SIM}_D$ s.t.

$$\Pr_{S \sim D^n}[\mathcal{A}(S) \overset{(\varepsilon, \delta)}{=} \text{SIM}_D] \geq 1 - \beta$$

In (Ghazi et al., 2024), the authors show any sufficiently DP algorithm is automatically perfectly generalizing.

**Theorem D.7** ((Ghazi et al., 2024) (Theorem 31, rephrased))**.** *There exists a universal constant $c > 0$ such that if $\mathcal{A}$ is $T$-user $\left(\frac{c}{\sqrt{T \log(T)}}, \frac{c}{T}\right)$-DP, then $\mathcal{A}$ is $(.5, .5, .5)$-perfectly generalizing. Moreover $\text{SIM}_D$ can be taken to be the distribution $\mathcal{A}(\cdot, \cdot)$, taken over samples and internal randomness.*

We remark that as written Theorem 31 in (Ghazi et al., 2024) is only stated for item-level DP, but the result follows immediately from viewing the user-level DP guarantee as standard DP over input space $\mathcal{X}^{n/T}$. Combined with Theorem B.2, it is then elementary to move from DP to global stability:

*Proof of Theorem D.1 and Theorem D.2 (Item 2).* The proof is exactly the same for Theorem D.1 and Theorem D.2, with the exception that we do not need to handle sample and confidence decay in the latter. We therefore argue just the former.

By assumption, we are given a $\beta'$-confident $(\varepsilon, \delta)$-DP algorithm $\mathcal{A}$ on $\ell = C_{DP}(n', \beta', \varepsilon, \delta)$ random bits and $n'$ samples whose privacy parameters meet the requirements of Theorem D.7. $\mathcal{A}$ is therefore $(.5, .5, .5)$-perfectly generalizing with

---

[8]Formally there may be some other measure 0 set of outputs given the exact statement of Theorem D.4, but these can be safely removed with no loss in parameters since we are in the approximate DP setting.

respect to $\mathcal{A}(\cdot, \cdot)$. Fix an input sample $S$ where $\mathcal{A}(S, \cdot)$ is $(.5, .5)$-close to $\mathcal{A}(\cdot, \cdot)$. Write $L = \mathrm{Supp}(\mathcal{A}(S, \cdot))$, and observe we have the trivial bound $|L| \leq 2^\ell$. On the other hand, distributional closeness implies

$$1 = \Pr[\mathcal{A}(S, \cdot) \in L] \leq e^{1/2} \Pr_{S \sim D^n, r}[\mathcal{A}(S, r) \in L] + \frac{1}{2},$$

and re-arranging, that

$$\Pr_{S \sim D^n, r}[\mathcal{A}(S, r) \in L] \geq \frac{1}{2e^{1/2}}.$$

Thus $\mathcal{A}$ has a $\frac{1}{2|L|e^{1/2}}$-heavy-hitter. We may now apply our parametrized heavy-hitter to globally stable conversion (Corollary B.3) to convert $\mathcal{A}$ to a $\beta'$-confident deterministic globally stable algorithm on $n = n' \cdot \tilde{O}(\frac{\log(1/\beta)}{|L|^2})$ samples for $\beta \leq O(\beta'|L|)$ as desired. $\qquad\square$

## E. The Stable Complexity of Agnostic Learning

In this section we bound the certificate complexity of PAC-Learning, resolving a main open question of (Chase et al., 2024) (stated in the equivalent language of global-stability/list-replicability). We restate the theorem here for convenience.

**Theorem E.1** (Certificate Complexity of Agnostic Learning). *Let $(X, H)$ be a hypothesis class of Littlestone dimension $d$. There exists a $\beta$-confident better than $\frac{1}{2}$-replicable learner with*

1. *Sample Complexity: $m(\alpha, \beta) \leq \exp(\mathrm{poly}(d)) \mathrm{poly}(\alpha, \log(1/\beta))$*

2. *Certificate Complexity: $C_{Rep}(\alpha) \leq \mathrm{poly}(d) + O(VC(H) \log \frac{1}{\alpha})$*

*Moreover, if $d = \infty$, $C_{Rep}(\alpha) = \infty$ for every $\alpha < \frac{1}{2}$.*

Our proof is inspired by (Bun et al., 2023)'s version of (Hopkins et al., 2022)'s realizable-to-agnostic reduction for replicable learners. (Bun et al., 2023) cannot be used directly due to its substantial reliance on additional randomness (indeed they actually have randoness scaling not only with confidence $\beta$, but with the size of the concept class $H$ which may be infinite in our case). Nevertheless, we build off the core idea, which uses the following realizable-case 'list-stable' learner of (Ghazi et al., 2021a;b):[9]

**Theorem E.2** (Realizable List-Stable Learning (Ghazi et al., 2021a;b)). *Let $(X, H)$ be a hypothesis class of Littlestone dimension $d$. There exists a algorithm on $n(\alpha, \beta) \leq \mathrm{poly}(d, \alpha, \log(1/\beta))$ samples which outputs a list of $\alpha$-accurate hypotheses of size at most $\exp(\mathrm{poly}(d))\alpha^{-O(1)}$ with probability at least $1 - \beta$. Moreover, for every distribution $D$, the list has a $\Omega(1/d)$-heavy-hitter, i.e. some hypothesis $h_D$ such that*

$$\Pr_S[h_D \in \mathcal{A}(S)] \geq \Omega\left(\frac{1}{d}\right).$$

Given the above, the core idea is to run this list-stable learner over all possible labelings of a large enough unlabeled data sample and return a commonly appearing element with high accuracy over the agnostic distribution. This results in a randomized learner with a good heavy hitter, which can then be converted to a globally stable learner by Theorem B.2 and finally a replicable learner with good certificate complexity by Theorem C.1.

*Proof of Theorem E.1.* Fix $\beta' = \exp(\mathrm{poly}(d))\alpha^{-O(1)}$ to be sufficiently smaller than the list size from Theorem E.2 and let $\mathcal{A}$ be the corresponding promised learner. Consider the following process generating a list of potential hypotheses from $\mathcal{A}$:

1. Draw $T = \mathrm{poly}(d, \log(1/\beta))$ unlabeled samples of size $n(\alpha/8, \beta')$ from $D$, denoted $S_U^{(i)}$.

2. Run $\mathcal{A}$ over all labelings of each $S_U^{(i)}$ in the class and denote the resulting hypothesis (multi)set as

$$C(S_U) := \{\mathcal{A}(S_U^{(i)}, h(S_U^{(i)})) \ : \ h \in H, i \in [T]\}$$

---

[9]We remark this is not quite as stated in (Ghazi et al., 2021b), which only claims exponential dependence on $\alpha$. However, polynomial dependence as stated here is immediate from (Ghazi et al., 2021a)

Note that by Lemma A.13, the size of $C(S_U)$ as a multiset is at most

$$|C(S_U)| \leq Tn(\alpha/8, \beta')^{O(VC(H))} \exp(\text{poly}(d)) \leq T \text{poly}(d, \alpha^{-O(1)})^{VC(H)} \exp(\text{poly}(d)).$$

The idea is now that $C(S_U)$ should contain many copies of a heavy hitter from the output list of some optimal labeling $h_{OPT}$ (which furthermore must be $\alpha/2$-correct by our confidence assumption). The only issues are 1) $C(S_U)$ may also contain many bad hypotheses, and 2) $|C(S_U)|$ currently depends on $T$ (which depends on the confidence $\beta$), so is too large for our purposes. We can fix both of these problems via a pruning step that removes any high error or non-heavy-hitter hypotheses. Denote by $w(h)$ the number of times $h$ appears in $C(S_U)$, and consider the following procedure:

1. Draw a new labeled sample $S_L$ of $\frac{\log |C(S_U)| + \log \frac{1}{\beta}}{\alpha^2}$ samples

2. Remove any hypothesis $h \in C(S_U)$ with high error or low empirical probability:

$$\text{Pruned}(C(S_U)) := \left\{ h \in C(S_U) : err_{S_L}(h) \leq \min_{h' \in C(S_U)} err_{S_L}(h') + \frac{3}{4}\alpha \text{ and } w(h) \geq c'\frac{T}{d} \right\}$$

3. Output a random hypothesis in $\text{Pruned}(C(S_U))$

where $c' > 0$ is some constant to be chosen. Note that now, by construction, $\text{Pruned}(C(S_U))$ has at most $\text{poly}(d/\alpha)^{VC(H)} \exp(\text{poly}(d))$ hypotheses since at most this many can appear $\Omega(T/d)$ times.

We now argue correctness and (heavy-hitter) global stability of the procedure (certificate complexity will then follow from Theorem B.2 and Theorem C.1). By standard uniform convergence bounds, we have that with probability at least $1 - \beta/2$ the empirical and true error of every $h \in C(S_U)$ are close:

$$|err_D(h) - err_{S_L}(h)| \leq \frac{\alpha}{8}.$$

Conditioned on the above, correctness follows as long as $C(S_U)$ actually contains a good hypothesis:

$$\min_{h' \in C(S_U)} err_{S_L}(h') \leq OPT_D + \frac{\alpha}{8}$$

since pruning then only keeps hypotheses with empirical error at most $\frac{7}{8}\alpha$ and therefore of true error at most $\alpha$ as desired.

In fact we will argue not just that this event occurs with probability at least $1 - \beta/2$, but that there exists a *fixed* $\alpha/8$-optimal hypothesis $h_D$ that appears with this probability. Since our final algorithm outputs a random hypothesis in $\text{Pruned}(C(S_U))$, this implies both our desired accuracy and global stability. To see this, fix some optimal hypothesis $h_{OPT}$ in the class, and observe that the original set $C(S_U)$ contains $T$ independent runs of $\mathcal{A}$ on samples labeled by $h_{OPT}$ (since we run over all labelings). $\mathcal{A}$ is promised to have an $\alpha/8$-optimal $\Omega(1/d)$-heavy-hitter $h_D$ with respect to the distribution $D_X \times h_{OPT}$ where the marginal over data is given by $D$ and the labeling is given by $h_{OPT}$. As a result $h_D$ itself is $\frac{1}{8}\alpha$-optimal with respect to the true distribution $D$. Moreover, since $T = \text{poly}(d, \log(1/\beta))$, a standard Chernoff bound implies $h_D$ appears at least $c'T/d$ times with probability at least $1 - \beta/2$ for the appropriate choice of constant $c' > 0$, completing the argument.

The above analysis promises the given algorithm is $\beta$-correct and has an $\Omega(\exp(\text{poly}(d))\alpha^{-O(1)})$-heavy hitter. We can convert this into the desired replicable learner using Theorem B.2 and Theorem C.1 with sample overhead at worst polynomial in the stability parameter as desired.

It is left to show that when $d = \infty$, there is no replicable (equivalently, globally stable) learner with better than $\frac{1}{2}$ error. To see this, first observe that a replicable agnostic learner with better than $\frac{1}{2}$ error implies a replicable learner in the realizable case with better than $\frac{1}{2}$ error. By (Impagliazzo et al., 2022), this can be boosted to a replicable realizable learner with arbitrary error $\alpha$, then transformed to a differentially private one by standard stability-to-privacy reductions (e.g. (Bun et al., 2020) or the transforms presented in this work). The existence of this final learner then violates the impossibility of private realizable learning of infinite Littlestone classes (Alon et al., 2019), completing the proof. $\square$

# F. Amplifying Replicability

A particularly useful property of replicability is that it can be easily amplified. In the main body, we implicitly gave a randomness-efficient procedure to amplify better than $\frac{1}{2}$ replicability to any $\rho$ based on converting to and from global stability. Here, we give a randomness inefficient but sample-efficient transform with the added benefit of starting from any starting replicability parameter $\nu$. The transform is similar to that in (Kalavasis et al., 2023) for so-called 'TV-indistinguishable' algorithms, but has improved sample complexity.

**Lemma F.1** (Generic Amplification of Replicability). *Fix $\nu, \beta > 0$, and let $\mathcal{A}$ be an $(1-\nu)$-replicable, $\beta$-correct algorithm on $n = n(\nu, \beta)$ samples. For any $\rho > 0$, there exists an efficient blackbox procedure amplifying $\mathcal{A}$ to a $\rho$-replicable, $\beta'$-correct algorithm $A'$ on $\tilde{O}\left(n \cdot \frac{\log(1/\beta)}{\rho^2\nu^2}\right)$ samples for $\beta' \leq \tilde{O}\left(\beta\frac{\log(1/\beta)}{\rho^2\nu^3}\right)$.*

*Proof.* Since $\mathcal{A}$ is $(1-\nu)$-replicable, by Markov's inequality there must be at least a $\nu/2$-fraction of $\mathcal{A}$'s random strings for which the distribution $\mathcal{A}; r)$ has an $O(\nu/2)$-heavy-hitter. Draw a set $T$ of $O\left(\frac{\log(1/\rho)}{\nu}\right)$ random strings. The probability that at least one drawn string has a $O(\nu/2)$-heavy-hitter is at least $1 - (1-\nu/2)^{O(\log(1/\rho)/\nu)} \geq 1 - \rho/2$.

Condition on this event and consider the algorithm which, given a sample $S$, outputs a list containing the result of $\mathcal{A}$ on $S$ across all random strings in $T$:
$$M(S) := \{\mathcal{A}(S; r) : r \in T\}$$

By assumption, $M$ satisfies the following properties, together called "list-stability" (Ghazi et al., 2021b; Bun et al., 2023):

1. $M(S)$ Outputs a list of size $|T|$

2. $\exists h$ s.t. $h \in M(S)$ with probability at least $O(\nu)$.

3. With probability at least $\beta' = |T|\beta$, all $h \in T$ are correct.

We now appeal to Theorem 6.7 in (Bun et al., 2023), which gives a $\rho/2$-replicable procedure that runs $M$ on

$$m = \tilde{O}\left(\frac{\log(1/\beta)}{\rho^2\nu^2}\right)$$

independent size-$n$ samples,[10] and outputs some hypothesis lying in one the resulting $M(S)$ with probability at least $1 - \beta$. By a union bound, all such hypotheses are correct except with probability $|T|m\beta$, and the entire procedure is $\rho$-replicable as desired. □

---

[10]Here $\tilde{O}$ hides (at most cubic) logarithmic factors in $\rho$ and $\nu$.

