# OpenReview forum: "The Role of Randomness in Stability"
_ICML.cc/2025/Conference — ICML 2025 spotlightposter_

### Official Review · Reviewer_3UmF · 2025-02-19

**Overall Recommendation:** 3

**Summary:**

This paper studies the notion of stability, which is defined (through various definition) as the probability of an algorithm to give the same result on two datasets sampled from the data distribution. More precisely, the authors aim at quantifying the amount of randomness needed to achieve stability, as randomness has been commonly seen as an important factor for stability.
The amount of randomness is considered in two settings (replicability and differential complexity) through two notions of "randomness complexity", namely certificate complexity and DP complexity, which have been introduced in prior works.
The two main results are to provide tight guarantees of these two notions of randomness complexity in terms of the globally-stable complexity, which quantifies the probability of a deterministic algorithm to output replicable results.
The authors then apply their framework to PAC-learning, for which they characterise the certificate complexity and the sample complexity of an algorithm achieving at least $\frac1{2}$-replicability.

**Claims And Evidence:**

The main claims are (i) tight bounds of certificate complexity and DP complexity in terms of globally-stable complexity and (ii) a characterisation of the certificate complexity of PAC learning in terms of the Littlestone dimension.

These claims are supported by proofs in the appendix and sketches of proofs are provided, which improve the readability of the paper.

However, several concepts such as Littleton dimension or share random bits are not defined formally in the paper (please correct me if I am wrong). As I am not an expert in this literature, it makes the statement look a bit informal and it is hard to assess their correctness.

**Essential References Not Discussed:**

I think a comparison with the literature of algorithmic stability (eg, Bousquet & Elisseeff, 2002) could be beneficial, as it is also quantifying the similarity of the output of the algorithm on similar dataset. Algorithmic stability is a major tool in the modern study of generalization error in learning theory and it would be beneficial to discuss it quickly, if it is relevant.

**Experimental Designs Or Analyses:**

N/A

**Methods And Evaluation Criteria:**

The proposed proof strategies, based on boosting theorems for stability, make sense for the problem at hand.

Most result are rather abstract, the clarity could be greatly improved by providing an instantiation of the results on a more practical/explicit example.

**Other Comments Or Suggestions:**

N/A

**Other Strengths And Weaknesses:**

The writing could be largely improved in my opinion, here are a few comments:

 - the first two paragraphs of the abstract and of the introduction are almost copy past of each other.
 - The abstract is very long (half a page), reducing it would improve clarity
 - Some citations, especially in the abstract, are not written using the standard bibliographic tools in latex but instead written in plain text in a format that is not consistent with the other citations. It makes it look like the paper is not finished.
 - some formulations are a bit informal, e.g., "taking a minute to explain",  "we'd also like", "an astronomical number of"
 - the paper has no conclusion
 - the section "further related works" seems to repeat several things already discussed in the introduction, in general a small part of the paper is dedicated to the presentation of the main result. I believe it would improve the clarity to compress a bit the related works to include more technical background on the mathematical setup.

**Questions For Authors:**

1. In the certificate complexity bound of theorem 2.4, which term is typically dominating in practice?

2. The results, while interesting, are presented in a quite abstract way, could it be possible to apply them on more explicit examples, like a specific classification setting for PAC-Learning?

3. Are there formal links between your stability notions and the algorithmic stability that is classically studied in learning theory.

**Relation To Broader Scientific Literature:**

This paper is strongly related to existing literature on stability, certificate complexity and DP complexity, which have all been relatively recently introduced.

**Theoretical Claims:**

The authors provide in Section 4 an overview of the technical elements involved in their proofs. However, I did not check the proofs in the appendix carefully, as it is a bit far from my expertise.

I find it confusing that several concepts, such as Littleton dimension, are not formally define in the text. I makes it hard to check the results and to get intuition about them. Adding a definition in the appendix could improve the clarity for readers that are not used to these notions.

I had one general remark about the framework: The complexity notions are defined via statements of the form $\mathrm{Pr}(\mathcal{A}(S) = \mathcal{A}(S')) \geq \eta$. Does this kind of definition implicitly assume that $\mathcal{A}$ takes values in a finite or countable space. For instance, if the hypothesis class is $R^d$ (eg, for stochastic optimizers such as SGD), then $\mathcal{A}$ could have a continuous PDF and the previous probability would always be 0. In general, I think the introduction of these notions could be made more formal (the notion of shared random bits is not clearly introduced in my opinion).

Minor remark: the notation $\mathcal{X}^\star$ is not defined.

At line 156 (second column), shouldn't the denominator be $\eta^3 \epsilon$?

---

> ### Author Rebuttal · Authors · 2025-03-30
>
> We thank the reviewer for their careful reading of our work, comments, and questions. Below we clarify a few possible misunderstandings about our definitions, explain our writing choices, and discuss several minor modifications we will make to address the reviewers concerns:
>
> **Q1: Missing Definitions** We respectfully disagree that there are substantial missing definitions in the paper (outside of Littlestone dimension which we chose to exclude for reasons explained below, but will include in the next version).
>
> The term “shared randomness” refers informally to the random string $r$ in the definition of replicability — it is not a standalone definition. Replicability promises that over independent samples $S$ and $S’$ and the *same random string* r:
>
> $Pr_{r,S,S’}[A(S;r)=A(S’;r)] \geq 1-\rho$
>
> $r$ is usually called the “shared randomness” since it is used in both runs of A. This is the standard way to express this notion in the literature — it is not usually defined on its own. There is, however, a confusing typo after this equation regarding the domain of $r$ for which we sincerely apologize. The sentence “the smallest $\ell$ s.t. there exists a better than $\frac{1}{2}$-replicable algorithm solving $\mathcal{M}$” should say “solving $\mathcal{M}$ *using $\ell$ random bits*”. We will fix this and clarify the domain in the text.
>
> *Regarding Littlestone:* the connection between Littlestone dimension and stability is very involved and largely orthogonal to our work. We do not use the definition in any of our proofs, so excluding it should not make any of our results harder to check. We use as a blackbox that any finite Littlestone class has a globally stable realizable learner (see Thm E.2). Our contribution thus has very little to do with Littlestone dimension, and we thought including the definition would only confuse readers as its connection to stability is fairly mysterious without deep knowledge of prior work.
>
> This said, we absolutely see the reviewer’s counterpoint that stating the definition and (perhaps more importantly) giving concrete examples would help the unfamiliar reader understand what types of classes Thm 2.4 applies to. We will include these in the text (see Q2 below) as well as add references to prior work discussing the above in more detail.
>
> **Q2: Concrete Examples** There are many nice examples of classes with finite Littlestone dimension to which Thm 2.4 applies. Two classical examples are affine subspaces in $\mathbb{R}^n$ (or more generally algebraic varieties), and half-spaces with margin (semialgebraic varieties with margin). Thm 2.4 gives the first randomness-efficient stable learners for both of these families. We will add discussion of this in the text.
>
> **Q3: Thm 2.4, Dominating Term**. This depends on how good accuracy you want, but if one only wants (say) 99% accuracy over the data, the first term will always dominate (e.g. this term would be polynomial in the dimension of the affine subspace).
>
> **Q4: Continuous Output Spaces** we do not make any assumptions about the output domain of $\mathcal{A}$. However, it is true that if the output distribution of $\mathcal{A}(\cdot)$ is continuous, then $\mathcal{A}$ is not globally stable for any $\eta>0$. The definitions we consider measure the performance of the *best* algorithm over the domain/output, so this is not an issue for working over uncountable and non-discrete domains.
>
> **Q5: Relation to [BE02]** Replicability and Differential Privacy are only loosely related to the classic stability notions of [BE02], which are generally much weaker (e.g. these notions can be satisfied for any VC class, whereas DP and replicability cannot). It is true there are some connections: e.g. replicability implies generalization by a similar Hoeffding-style argument, and uniform stability [BE02] is related to a substantial relaxation of DP known as private prediction. To the best of our knowledge, however, there is no real formal comparison between [BE02] and the models we study, though the work is certainly worth mentioning as a seminal investigation into stability in learning.
>
> **Other Writing Comments**: Our citation style in the abstract is a conscious choice ensuring all authors of the major relevant papers are listed (the ICML citation format does not seem to allow this directly). This is because theory papers have alphabetical author order and all authors should be credited equally. We also note it is relatively common for theory papers to have a “discussion/open-problem” section as we do rather than a traditional conclusion and respectfully disagree this is an indicator of poor writing.
>
> This said, we do agree the abstract is on the long side for the ICML format and repetitive with the intro, so we will shorten it as the reviewer suggests and ensure the early definitions are as clear as possible.

---

> > ### Comment · Reviewer_3UmF · 2025-04-02
> >
> > Thank you for taking the time to clarify several points.
> >
> > I have now a better understanding of how the paper is written and I believe that the proposed changes will improve the clarity of your work.
> > For this reason, I will increase my score by one point.
> >
> > Good luck!

---

### Official Review · Reviewer_jLTx · 2025-03-14

**Overall Recommendation:** 4

**Summary:**

This paper addresses the randomness complexity of an algorithm: how many random bits an algorithm requires to satisfy certain property. Two properties are studied: replicability and differential privacy. It is shown that the randomness complexities of these properties are closely connected to the global stability of the algorithm. The latter can be formulated as a property of a deterministic algorithm. Two main results follow from the boosting from deterministic globally-stable algorithms to replicable or private (necessarily) randomized algorithms.

**Claims And Evidence:**

Yes

**Essential References Not Discussed:**

Not to the best of my knowledge.

**Experimental Designs Or Analyses:**

Not applicable

**Methods And Evaluation Criteria:**

Yes

**Other Comments Or Suggestions:**

Line 774: should be $C_{\mathrm{Glob}} \leq \log \frac{1}{\eta}$

**Other Strengths And Weaknesses:**

The paper is well-written and provides an important contribution by showing a sharp relation between fundamental algorithmic properties.

A weakness: Theorem 2.3 only considers the high-privacy regime ($\varepsilon = O(1 / \sqrt{n' \log n'})$).

**Questions For Authors:**

Please clarify the end of the proof of Theorem B.2. For example, what is $L$ in line 771?

**Relation To Broader Scientific Literature:**

This work contributes to the recent line of research on randomness complexity of stable algorithms. While prior results (e.g. Canonne et al., 2024) look at particular algorithmic tasks, this work provides a general equivalence between stability properties, such as global stability, replicability and differential privacy.

**Theoretical Claims:**

I checked results in Appendix B in detail. Modulo clarity of expositions, the theoretical claims are correct.

---

> ### Author Rebuttal · Authors · 2025-03-29
>
> We thank the reviewer for their careful reading of our work and helpful comments. We will make the suggested minor fixes.
>
> *Regarding the end of the proof of Thm B.2:*
>
> The $L$ in line 771 is a typo and should be $L_D$, our apologies. The point here is to bound the measure of the set of samples $S$ on which $A_{\text{List}}$'s majority output is not in $L_D$; in particular, we want to show this bad set has measure at most $\frac{2\gamma}{\eta}$.
>
> To see this, we observe that for any *fixed* sample S whose majority output is not in $L_D$, it must be the case that $A_{\text{List}}(S;\cdot)$, over the randomness of $r$, outputs a hypothesis *outside of* $L_D$ with probability at least $\eta/2$. But then if there is a $\frac{2\gamma}{\eta}$ measure of such samples S, the overall probability that $A_{\text{List}}$ outputs something outside of $L_D$ (now over the randomness of both S and r) is more than $\gamma$. This contradicts $A_{\text{List}}$’s list-replicability (the guarantee on Line 752) and completes the proof.

---

### Official Review · Reviewer_6PrE · 2025-03-17

**Overall Recommendation:** 4

**Summary:**

The authors study the randomness complexity of private and replicable algorithms, showing that this complexity is tightly related to the best global stability parameters of an algorithm for the same task. Letting $\eta_M$ be the best achievable replication probability for a deterministic algorithm solving problem $M$, they define the globally-stable complexity of a problem, $C_{Glob} = \log \tfrac{1}{\eta}$. They then show that the certificate complexity of $M$ is lower-bounded by $C_{Glob}$ and upper-bounded by $C_{Glob} + 1$. The similarly give upper and lower bounds on the DP complexity of a problem $M$ by a function  of $C_{Glob}$ and privacy parameters $\varepsilon, \delta$. Finally, they show that the certificate complexity of agnostic PAC learning for classes with finite Littlestone dimension (ensuring they are replicably learnable) is $poly(d) + O(VC(H)\log(1/\alpha))$. Thus, the certificate complexity for agnostic PAC learning resembles that of realizable learning, up to O(VC(H)) factors.

**Claims And Evidence:**

All claims are supported with proof.

**Essential References Not Discussed:**

All essential references are discussed.

**Experimental Designs Or Analyses:**

N/A

**Methods And Evaluation Criteria:**

Yes.

**Other Comments Or Suggestions:**

Page 3 - Double close parens in the first paragraph after Theorem 2.1

In the text preceding Theorem 2.3, I was a bit confused about what happened to the confidence parameter in moving from DP to user-level DP, until I got to the appendix.

Page 4 - “Not only is such a bound is possible”

Page 7 - “occurring probability”

**Other Strengths And Weaknesses:**

The paper was very well-written and a pleasure to read. The results represent a significant contribution to our understanding of the randomness complexity of stable algorithms.

One weakness (acknowledged by the authors) is the fairly significant tradeoff between sample and randomness efficiency in the results. The randomness complexity is likely to be a secondary or tertiary consideration in most applications, after sample and computational complexity, so understanding the randomness complexity of sample and/or computationally efficient stable algorithms seems like the kind of result we’d really love to have. This is of course a much more significant ask (particularly the computational efficiency piece).

**Questions For Authors:**

No questions for the authors.

**Relation To Broader Scientific Literature:**

The authors do a very thorough job of connecting their results to the existing literature, highlighting how their work extends prior work on DP complexity, certificate complexity, and answers open questions regarding agnostic learning raised in prior work.

**Theoretical Claims:**

I did not go carefully through the proofs in the appendices, but the explanation of proof approach is clear enough that they imply the stated results.

---

> ### Author Rebuttal · Authors · 2025-03-28
>
> We thank the reviewer for their careful reading of our work and helpful comments (we will fix the minor issues noted).
>
> The trade-off between sample and randomness efficiency is a very interesting question, and is not well understood even for the most basic examples (see e.g. “Geometry of Rounding: Near Optimal Bounds and a New Neighborhood Sperner's Lemma” and “The Randomness Complexity of Differential Privacy”). Our transformations have an overhead scaling polynomially in the stability parameter, which can be efficient for low-dimensional problems but is indeed costly in settings like PAC learning. In the PAC case, this can be circumvented when one does not care about randomness by looking at certain variants of global stability, and it would be interesting to understand if there is an inherent trade-off here. However, given the substantial difficulty in resolving this problem even for, say, estimating the bias of d coins, we feel this is reasonably outside the scope of our work.

---

### Official Review · Reviewer_poww · 2025-03-20

**Overall Recommendation:** 4

**Summary:**

This paper studies a type of stability for algorithms and its connections to differential privacy (DP). It is argued that such stability requires randomization of the algorithm, and results are given for translating between different measures of complexity. The theoretical results are explained with intuition, and proof sketches are given in the main text with complete proofs in the appendices.

I am not familiar with this field, so my lower score mostly reflects a lack of confidence.

**Claims And Evidence:**

Most claims are theoretical; see below.

**Essential References Not Discussed:**

N/A

**Experimental Designs Or Analyses:**

The authors did not perform any numerical experiments.

**Methods And Evaluation Criteria:**

N/A

**Other Comments Or Suggestions:**

* $\rho$-replicable definition: the distribution for $r$ is never stated
* line 084 col 2: $\ell$ is never defined
* line 088 col 2: "$\ell$-random" should be "$\ell$ random"
* explicit formulae for $C_\mathrm{Rep}$ and $C_\mathrm{Glob}$ should be given in the main text
* Thm 2.2: In item 2 (DP to stability) it seems like the prime on $\beta$ is swapped from where it should be; is this a typo?
* line 240 col 2: "tower-type dependence" this wasn't familiar to me, you might want to explain what you mean by this
* line 348 col 1: there is a typo with the first argument missing from $\mathcal{A}_\mathrm{Rep}(;r)$

**Other Strengths And Weaknesses:**

The claim that "stability," in the colloquial sense, is important for algorithms seems self-evident. However, I have a philosophical quibble with the form of stability that the authors study here. My background is applied mathematics and I'm familiar with the type of stability studied in numerical analysis where the output of an algorithm is required to be close for close inputs. In the current work, the authors require the output to be the *same* for different inputs, which seems a very strong requirement. It would help convince people with backgrounds like mine if the authors would motivate better why they've chosen such a strong type of stability. In my opinion, requiring the exact same outputs for different datasets is more than anyone practically would want, but I'm willing to admit I could be wrong.

**Questions For Authors:**

* Can you better motivate the particular definition of stability for people who are not familiar with this area?
* Could you shorten the abstract? It is very long.

**Relation To Broader Scientific Literature:**

I am not familiar with this literature, so I don't have much to add. However, it would be good for the authors to address other kinds of stability, e.g. that used in numerical analysis and its differences from the current form in the introduction.

**Theoretical Claims:**

Overall, the results sound fairly reasonable to someone who isn't an expert. I did not check the correctness of the proofs in the appendices.

A few things stood out to me, however:
* The idea that randomness is required for stability is not self-evident, even after reading footnote 1. In many cases data are discrete (say inputs are binary strings) so I don't understand how an interpolation argument like that given here would apply.
* Thm 2.2 gives a result in terms of failure probablity $\beta$ versus $\beta'$. Since the confidence takes the form of a probablity $1-\beta$ (line 071), it would seem to me that your results could easily become vacuous if that grows exponentially as in Thm 2.2 (parts 1 & 2). In other words, the blowup in lack of confidence doesn't seem so mild to me.

---

> ### Author Rebuttal · Authors · 2025-03-29
>
> We thank the reviewer for their careful reading and questions and will make all suggested fixes
>
> We hope the below clarifications (and positive impressions of the area expert reviewers) raises the reviewer’s confidence in our work, and respectfully ask they consider using the confidence score rather than a lower main score to reflect any remaining lack of confidence due to field familiarity.
>
> **Q1:** “The idea that randomness is required for stability is not self-evident.  I don't understand how an interpolation argument would apply.”
>
> **A1:** Good catch. The footnote only formally makes sense for DP (where “similar” means trading out one example in your discrete data-set, so one can always interpolate by trading out points one by one). In replicability, the argument is similar but one interpolates in *parameter/distribution-space*, not over samples. We will clarify this in the text.
>
> For example, consider estimating the bias of a coin. The data is discrete (coin flips), but the underlying set of distributions (biases $p \in [0,1]$) is continuous. Our interpolation is over p. A deterministic replicable algorithm has a single canonical output w.h.p for every $p \in [0,1]$, but cannot distinguish between close p and p’. Since we can interpolate between any $p,q \in [0,1]$, the algorithm must have the same canonical solution everywhere so is essentially constant.
>
> **Q2:** “Thm 2.2 gives a result in terms of failure probability $\beta$ vs $\beta’$… the blowup in lack of confidence doesn't seem so mild to me”
>
> **A2:** This type of blowup is considered mild in the literature because almost all statistical problems have complexity scaling in $\log(1/\beta)$ due to Chernoff. This means one does not really lose significant generality assuming the initial algorithm has small $\beta$. More generally, we can usually amplify a constant success algorithm to $1-\beta$ success just by repeating it $\log(1/\beta)$ on independent samples and taking the best output. We will clarify this in the text.
>
> It is also worth remembering $C_{\text{glob}}$ is log scale, so, combined with the above, the blowup we incur is only, e.g., $loglog(1/\eta)$ for $\eta$-stability, which is mild even when $\eta$ is exponential (as in PAC Learning).
>
> **Q3:** “Can you better motivate the particular definition of stability for people who are not familiar with this area?”
>
> **A3:** Absolutely. This critical question has been discussed extensively in prior work (e.g. in Impagliazzo et al.). We’ll summarize here and add discussion in the paper as well as on weaker stability notions. We’ll focus on replicability, since presumably DP (widely adopted in industry and hugely impactful in theory) does not need more motivation.
>
> Replicability comes with a substantial list of advantages due to requiring truly equal outputs. Here is a representative sample:
>
> 1) Replicability is *closely related to other core notions of stability in computer science* (DP, adaptive data analysis, and more, see e.g. “Stability is Stable: Connections between Replicability, Privacy, and Adaptive Generalization” or “The Bayesian Stability Zoo”). Several open problems in DP (sample-efficient user-level DP, amplification of DP) were resolved through studying replicability.
>
> 2) Replicability is *testable*. It is not always easy to test whether an algorithm is stable (e.g. testing DP is known to be computationally hard). By requiring equivalence, replicability is trivially verifiable.
>
> 3) Replicability is *preserved under arbitrary post-processing*. One can apply any desired algorithm to the output of a replicable procedure, and the result will always remain replicable. This is an extremely useful property in algorithm design.
>
> 4) Replicability is *easily amplified*. Given a constantly replicable algorithm, there is a simple and computationally efficient procedure to amplify it to arbitrarily high replication probability.
>
> 5) Replicability is *model independent*. Different statistical tasks naturally have different notions of weak stability and closeness, or, in the case of testing problems with binary outputs, may have *no* natural notion of weak stability (see “Replicability in High Dimensional Statistics” or “Replicable Uniformity Testing” for a discussion of replicability in these settings). Replicability gives a **unified definition** for studying stability for all statistical tasks.
>
> It is also critical to understand when replicability is possible, and its overhead compared to weaker notions of stability. This has been the topic of a substantial number of works, and efficient replicable algorithms are known for many tasks (statistical queries, hypothesis testing, mean estimation, RL, learning large margin halfspaces…). This is not the main focus of our work, but we feel combined with the above more than justifies the study of replicable algorithms.
>
> **Q4:** Could you shorten the abstract?
>
> **A4:** We’ll shorten the abstract in the next version to be less repetitive with the intro.

---

> > ### Comment · Reviewer_poww · 2025-04-02
> >
> > Re: Q1
> > Your argument makes more sense to me now, but if the only replicable algorithm for estimating coin bias is "essentially constant", then is that definition of stability useful? I think what you would want, in practice, is a definition that allows for slightly different estimates of the bias $p$ for slightly different input data. In any case, this kind of issue is what I think you should discuss more when motivating the definition (when you address Q3).
> >
> > Reviewer 3UmF also had issues with the clarity of the random bit distributions (these need to be fully defined in the main text).
> >
> > Also, I'd like to hear your responses to my "other comments" section to be sure I was interpreting things right.
> >
> > Unfortunately year's review for ICML doesn't allow for confidence scores, only a single score. If you can clarify how you'll address the above comments, I'll be willing to raise my score 1 point.

---

> > > ### Author Response · Authors · 2025-04-05
> > >
> > > Thanks for the reply -- we ran out of space above (new ICML rebuttal limit), so can finish responding to your comments here.
> > >
> > > **Re Q3 and motivation:** We will definitely include a discussion of the advantages and disadvantages of replicability compared to other stability notions in the text. The notion suggested by the reviewer for the coin problem is natural and worth studying, but it turns out in this setting the stronger notion of replicability is not only possible, it's *asymptotically free!* We apologize if our previous explanation regarding the coin problem was confusing here — the interpolation argument is only against **deterministic** replicable algorithms. There is an elementary *randomized* (say) $\frac{3}{4}$-replicable algorithm for the coin problem running in time $O(\log(1/\beta)/\alpha^2)$ (for $\alpha$-accuracy and $\beta$-confidence), the same asymptotic cost as standard bias estimation. This was one of the initial motivations of Impagliazzo et al. [ILPS22] in defining the notion. As we mentioned before, efficient replicable algorithms are now known for a vastly larger class of problems than just bias estimation.
> > >
> > >
> > > That said, in general there are interesting tradeoffs between weak notions of stability like the one the reviewer suggests and strong notions like replicability. For instance, if one wants very high probability stability or is handling very high dimensional data, weak stability will likely be computationally and statistically cheaper. On the other hand, for settings where the output of the algorithm may then be post-processed by a highly sensitive function, replicability will preserve stability while weaker notions will not. This is the reason weak notions typically don’t lead to (differential) privacy; it’s possible one can use these small variations to extract private user data. The type of stability one should choose is based on the circumstances (there are of course many more reasons to use one vs the other, e.g. those described in our previous response), and developing a better understanding of both weak and strong stability can help us decide when to use which notion.
> > >
> > > **Response to “Other Comments” Section**:
> > >
> > > — We thank the reviewer for catching our fairly embarrassing typo regarding $\ell$. The sentence should read “the smallest $\ell \in \mathbb{N}$ s.t. there exists a better than $\frac{1}{2}$-replicable alg solving $\mathcal{M}$ *using $\ell$ random bits*”. In other words, a replicable algorithm *whose randomness $r$ is drawn uniformly from* {$0,1$}$^{\ell}$. We will fix this in the text.
> > >
> > > — *Regarding the domain of r.* We hope the above (at least partially) clarifies this. In the original definition of replicability the authors are not careful about specifying the domain of the randomness (usually one just assumes access to an infinite stream of Ber(1/2)-distributed bits the algorithm can query). In our setting (and that of Dixon et al.), the randomness is drawn uniformly from {$0,1$}$^{\ell}$ where $\ell \in \mathbb{N}$ may be taken arbitrarily large and can be viewed as a parameter of the algorithm. We will ensure this is formalized in the main text, along with the definitions of C_{glob} and C_{rep}
> > >
> > > We remark that technically, prior works even drew randomness from continuous domains such as uniform over [0,1], but these methods can easily be discretized to work over {$0,1$}$^{\ell}$ for large enough $\ell$.
> > >
> > > — The $\beta$ vs $\beta’$ in Thm 2.2 is a typo, thanks again. We’ll also fix the missing argument in 348 and typo at 088.
> > >
> > > — “Tower type dependence” refers to iterated exponentials, i.e. of the rough form $T(n) = 2^{T(n-1)}$, we’ll add this.

---

### Decision · Program_Chairs · 2025-05-01

**Decision:**

Accept (spotlight poster)

**Comment:**

The paper presents a randomness complexity theory for PAC learning under two popular notions of algorithmic stability: replicability and differential privacy. After discussions, the reviewers unanimously agreed that this is a theoretically solid paper that reveals some new and important connections between algorithmic stability, replicability and differential privacy. In their intial reviews, the revidwers also expressed some minor concerns regarding the significant trade-off between sample and randomness complexity in the results, as well as the writing quality which I agree still has room for improvement. The authors managed to address some of these in their thorough responses. Overall, this is a nice learning theory paper worth to be accepted for presentation at ICML 2025. The authors are encouraged to make the promised changes in the camera-ready version to address the reviewer's comments as much as possible.